# Proteogenomic characterization of 2002 human cancers reveals pan-cancer molecular subtypes and associated pathways

Yiqun Zhang [1,8], Fengju Chen[1,8], Darshan S. Chandrashekar[2,3,8], Sooryanarayana Varambally[2,3,4] & Chad J. Creighton [1,5,6,7 ✉]

Mass-spectrometry-based proteomic data on human tumors—combined with corresponding multi-omics data—present opportunities for systematic and pan-cancer proteogenomic analyses. Here, we assemble a compendium dataset of proteomics data of 2002 primary tumors from 14 cancer types and 17 studies. Protein expression of genes broadly correlates with corresponding mRNA levels or copy number alterations (CNAs) across tumors, but with notable exceptions. Based on unsupervised clustering, tumors separate into 11 distinct proteome-based subtypes spanning multiple tissue-based cancer types. Two subtypes are enriched for brain tumors, one subtype associating with MYC, Wnt, and Hippo pathways and high CNA burden, and another subtype associating with metabolic pathways and low CNA burden. Somatic alteration of genes in a pathway associates with higher pathway activity as inferred by proteome or transcriptome data. A substantial fraction of cancers shows high MYC pathway activity without *MYC* copy gain but with mutations in genes with noncanonical roles in MYC. Our proteogenomics survey reveals the interplay between genome and proteome across tumor lineages.

[1] Dan L. Duncan Comprehensive Cancer Center Division of Biostatistics, Baylor College of Medicine, Houston, TX, USA. [2] Comprehensive Cancer Center, University of Alabama at Birmingham, Birmingham, AL 35233, USA. [3] Division of Molecular and Cellular Pathology, Department of Pathology, University of Alabama at Birmingham, Birmingham, AL 35233, USA. [4] The Informatics Institute, University of Alabama at Birmingham, Birmingham, AL 35233, USA. [5] Department of Bioinformatics and Computational Biology, The University of Texas MD Anderson Cancer Center, Houston, TX, USA. [6] Human Genome Sequencing Center, Baylor College of Medicine, Houston, TX 77030, USA. [7] Department of Medicine, Baylor College of Medicine, Houston, TX, USA. [8] These authors contributed equally: Yiqun Zhang, Fengju Chen, Darshan S. Chandrashekar. ✉email: creighto@bcm.edu

The central dogma of molecular biology states that DNA makes RNA, and RNA makes protein. At the same time, many regulatory mechanisms occur after mRNAs are manufactured, and steady-state transcript abundances only partially predict protein abundances[1,2]. Recent technological advances in mass spectrometry have allowed for large-scale surveys of the cancer proteome. The Clinical Proteomic Tumor Analysis Consortium (CPTAC) is a National Cancer Institute initiative to accelerate the understanding of the molecular basis of cancer through the application of large-scale, mass spectrometry-based proteomics. CPTAC has carried out individual studies for a range of tissue-based cancer types[3–16], integrating proteome with genome, collectively known as proteogenomics, while efforts outside of CPTAC have similarly studied additional cancer types[17–19]. In these recent studies, proteogenomic analyses carried out within specific cancer types have included protein-level tumor versus normal adjacent tissue comparisons, cataloging the functional consequences of somatic mutation and copy number alteration (CNA) on the proteome, and defining tumor molecular subtypes and associated pathways and immune cell infiltrates.

CPTAC and other studies have collectively made mass spectrometry-based proteomic data on over 2000 human tumors available in the public domain, with corresponding data at the molecular levels of mRNA, copy number, and small mutation. These data present an opportunity for systematic and pan-cancer analyses of the entire cohort of tumors with proteomic data, including defining molecular subtypes and associated pathways relevant to multiple cancer types. A combined large pan-cancer cohort of tumor proteomic profiles would allow for proteogenomic analyses to identify commonalities, differences, and emergent themes across tumor lineages[20]. In recent studies, we carried out pan-cancer proteomic studies to define molecular subtypes[21] and protein correlates of more aggressive disease[22], each of these studies involving a more limited set of tumors and cancer types.

In this work, we comprehensively analyze proteomics data and corresponding multi-omics data on 2002 primary tumors from 14 different tissue-based cancer types. We explore protein-level versus corresponding mRNA-level associations, noting associations found in the transcriptome but not the proteome and vice versa. We define pan-cancer, proteome-based subtypes that cut across tumor lineages. Finally, we explore the interactions between the cancer proteome and somatic DNA-level alteration of cancer-associated pathways across tumors.

## Results

**Protein-level correlations**. We assembled a compendium dataset of mass spectrometry-based proteomics data of 2002 primary tumors from 17 individual studies[3–19] (Supplementary Data 1). The cancer types represented in this proteomic compendium dataset included breast ($n = 230$ tumors with proteomics data), colorectal ($n = 187$), gastric ($n = 80$), glioblastoma ($n = 100$), head and neck ($n = 108$), liver ($n = 165$), lung adenocarcinoma ($n = 111$), lung squamous ($n = 110$), ovarian ($n = 269$), pancreatic ($n = 137$), pediatric brain ($n = 219$), prostate ($n = 76$), renal ($n = 110$), and uterine ($n = 100$). For most of these tumors, corresponding multi-omics data were available for mRNA ($n = 1899$ out of the 2002 tumors), DNA somatic small mutation (single nucleotide variants (SNVs), and insertions/deletions, i.e., indels; $n = 1698$), and DNA somatic CNA ($n = 1837$). For the proteomic and transcriptomic compendium datasets, we normalized expression values within each cancer type, whereby neither tissue-dominant differences nor inter-laboratory batch effects would drive the downstream analyses[21,23–25]. The total proteomics compendium dataset consisted of 15,439 genes with proteins measured in at least one tumor, including 10,129 genes

with proteins represented in half of the tumors for at least seven cancer types profiled. The phospho-protein compendium consisted of 199,284 phospho-protein features involving 11,671 genes, 5419 phospho-proteins represented in half of the tumors for at least seven cancer types profiled. To facilitate access by biomedical researchers, we integrated the above proteomic data with the UALCAN data portal[21,22,26], allowing users to query proteins of interest for comparisons of interest (http://ualcan.path.uab.edu/).

Overall, protein expression of genes broadly correlated with the corresponding mRNA levels or CNA status across tumors, but with notable exceptions (Supplementary Data 2 and 3). Across 1899 tumors and 10,129 genes, tumors displayed a median gene-wise protein versus mRNA correlation $r$ value of 0.40 (Fig. 1a, b), with 97.1% of genes having significant positive correlations (Pearson correlation $p < 0.01$). For specific functional categories of genes, protein-mRNA correlations tended to be higher or lower. For example, mRNA levels of genes involved[27] in ribosome, oxidative phosphorylation, electron transport chain, and humoral immune response pathways tended to correlate poorly with protein expression across tumors (Fig. 1c), consistent with previous observations in individual cancer types[6,12,15]. Based on 9744 genes with available data, gene-level correlations between protein and CNA were broadly positive. However, protein versus CNA correlations tended to be lower than the corresponding mRNA versus CNA correlations (Fig. 1d). For the 13 cancer types examined, median gene-wise protein versus CNA Pearson correlation $r$ values ranged from 0.04 (prostate) to 0.21 (lung squamous) across the 9744 genes, while mRNA versus CNA correlation $r$ values ranged from 0.12 to 0.41 ($p < 1E{-}7$, paired $t$-test, comparing median gene-wise correlation by cancer type for protein versus mRNA). Taking a set of 756 genes with significant positive correlations between protein and CNA but not between mRNA and CNA ($p < 0.001$ and $p > 0.05$, respectively), these genes were enriched for many gene categories, including "protein-containing complex," "mitochondrial," "protein complex," "ribosomal subunit," "translation," "electron transport chain," "DNA repair," and "regulation of cell cycle process" (Fig. 1d). For individual genes of interest (e.g., genes involving the PI3K/AKT/mTOR pathway, Fig. 1e), copy loss and indel mutations could be associated with lower protein expression.

Proteomic signatures associated with higher tumor grade could yield molecular clues as to processes underlying more aggressive tumors. To identify proteomic correlates of grade, we followed an approach previously demonstrated in a limited dataset of 558 tumors from five cancer types[22]. In this present study, we found on the order of hundreds of total proteins and mRNAs differentially expressed with higher grade ($p < 0.01$, Pearson correlation) for each of ten cancer types (head and neck, liver, lung adenocarcinoma, lung squamous, ovarian, pancreas, pediatric glioma, prostate, renal, uterine), involving 1265 tumors for which grade information was available (Fig. S1a, b). Significant overlapping gene features were between the total protein signatures and the mRNA signatures (Fig. S1c). Each cancer type showed a proteomic and mRNA signature of grade distinctive from the other cancer types. At the same time, differential expression patterns involving a subset of genes were also shared across multiple cancer types (Fig. 2a), with 1936 proteins being significant ($p < 0.01$) for any two or more cancer types. Differentially expressed proteins and mRNAs were enriched for specific pathways[28] (Figs. 2b and S1d), with many pathways significant for multiple cancer types. The set of top enriched pathways uncovered here overlapped highly with the pathways we uncovered previously[22] based on five of the nine cancer types represented in our compendium. This finding suggests a core set of pathways associated with more aggressive

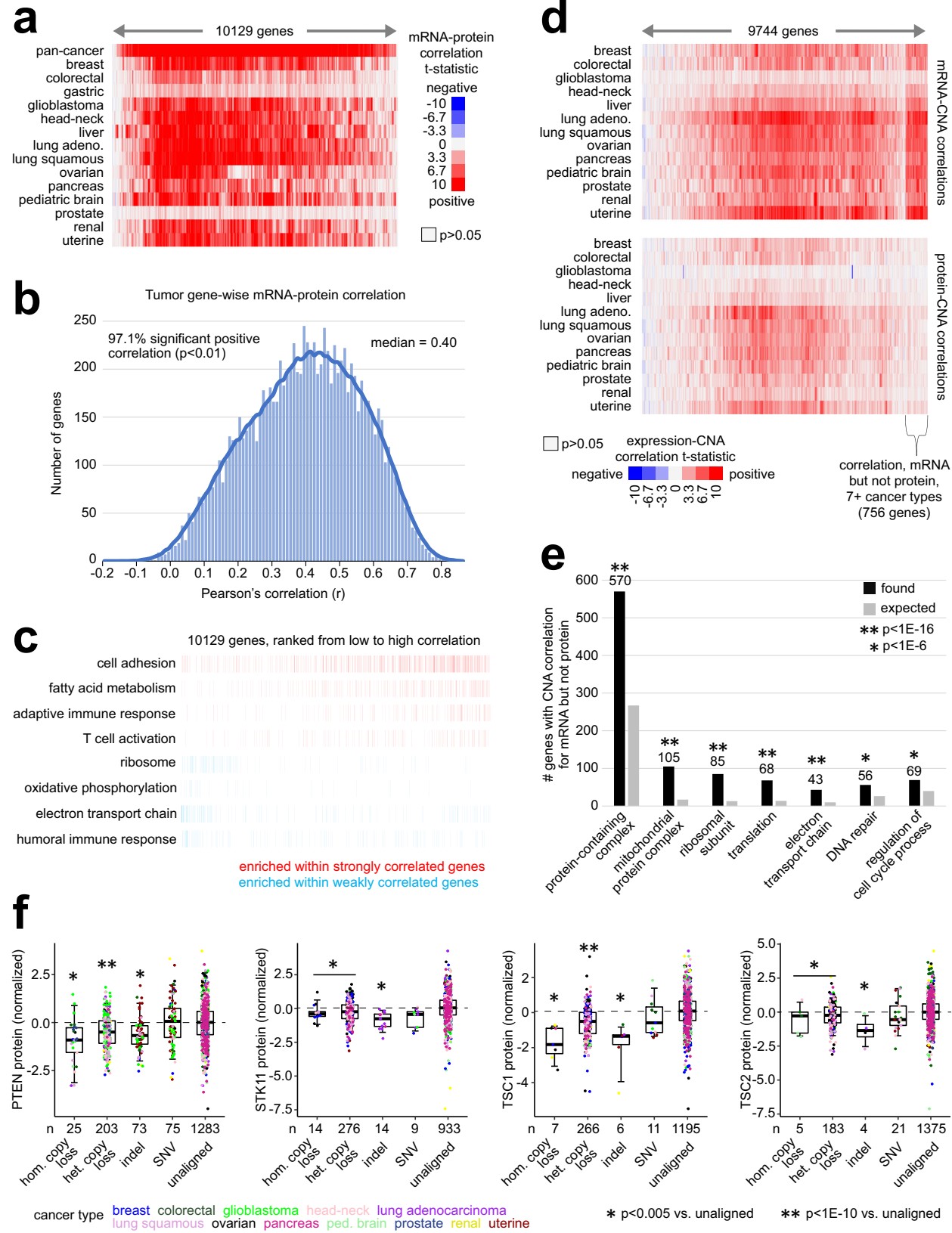

tumors, even as the proteomic grade correlates differ by cancer type. Additional pathways in this study include alpha 6 beta 4 signaling (liver, pancreas) and oxidative damage (prostate, uterine). One example pathway of interest enriched within

proteins increased with higher tumor grade, significant ($p < 0.05$, one-sided Fisher's exact test) for four out of ten cancer types—lung adenocarcinoma, ovarian, renal, and uterine—was type II interferon signaling (Fig. 2c), involving JAK–STAT-signaling and

**Fig. 1 Correlations of proteomic abundances with transcriptomics, copy number alterations, and somatic mutations. a** Heat map of gene-wise Pearson correlations of mRNA and protein expression (10,129 genes, with protein data for >6 individual cancer types) across all tumors studied ($n = 1899$ with available protein and mRNA data) as well by individual cancer type. Red, significant positive correlation. **b** Histogram of gene-wise Pearson correlations of mRNA and protein expression in 1899 tumors, based on 10,129 genes. **c** Annotated gene sets (by Gene Ontology, or GO) across 10,129 genes ranked by Pearson gene-wise correlation between protein and mRNA. Enrichment or anti-enrichment patterns are significant with $p < 0.0001$, FDR < 0.0001 by GSEA[71]. **d** Heat maps of gene-wise Pearson correlations of both mRNA expression and copy number alteration (CNA) and protein expression and CNA (9744 genes, out of the 10,129 with combined mRNA/CNA data for at least one cancer type), by individual cancer type. Red, significant positive correlation. **e** Out of 9744 genes, 756 were significantly positively correlated between expression and CNA for mRNA (Pearson $p < 0.001$) but not for protein ($p > 0.05$) for at least seven cancer types. Selected enriched GO terms for these genes are shown. $p$ values by one-sided Fisher's exact test. "Expected," estimated by probability. **f** Box plots of expression of selected proteins (PTEN, STK11, TSC1, TSC1) by somatic alteration class ("hom.," homozygous; "het.," heterozygous; "SNV," Single Nucleotide Variant; "indel," insertion/deletion). $p$ values by $t$-test. Box plots represent 5% (lower whisker), 25% (lower box), 50% (median), 75% (upper box), and 95% (upper whisker). $n$ = biologically independent tumors.

antiviral and growth-inhibitory effects. Another enriched pathway of interest—significant ($p < 0.05$) for liver, renal, and uterine cancer types—was oncostatin M cytokine signaling (Fig. 3d), known to play a significant role in inflammation, autoimmunity, and cancers[29].

**De novo proteome-based subtypes.** Molecular subtypes can provide insights into the pathways and processes appearing deregulated within tumor subsets[21,23]. The 2002 tumors in the proteomics compendium dataset separated into 11 distinct proteome-based pan-cancer subtypes based on unsupervised clustering (Fig. S2). These subtypes, s1 through s11, each spanned multiple tissue-based cancer types (Fig. 3a). Notably, the s11 subtype was specific to brain tumors, spanning glioblastomas and pediatric brain tumors. Previously, ten proteome-based pan-cancer subtypes, referred to as k1 through k10, were identified based on 532 tumors and five tissue-based cancer types[21]. Within these 532 tumors, there were significant patterns of overlap between the s1–s11 subtype assignments and the previous k1–k10 subtype assignments (Fig. 3a). Specifically, the correspondence between the current versus the previous subtyping included s1 to k1 (previously associated with the proteasome complex and ubiquitin[21]), s2 to k2 (associated with T cells and the immune response), s3 to k4 (associated with basal-like breast cancer), s4 to k5 (associated with epithelium and oxidative phosphorylation and TCA cycle pathways), s5 to k6 (associated with tumor stroma), s6 to k7 (associated with tumor stroma and collagen VI), s8 to k9 (associated with hemoglobin complex), and s9 to k10 (associated with endoplasmic reticulum and steroid biosynthesis pathway). The s7, s10, and s11 subtypes did not strongly correspond to the previous k1–k10 subtypes. The 11 proteome-based subtypes of the present study were each characterized by widespread differential expression patterns at the levels of both total protein and phospho-protein (Fig. 3b and Supplementary Data 4). Of the 1073 proteins for which total levels best distinguished the subtypes, 225 proteins had a drug target association by DrugBank[30] (Supplementary Data 4).

Previously compiled gene annotations[27] and gene signatures[21,23] helped characterize the proteome-based subtypes represented by the compendium dataset (Supplementary Data 5). Within the top differentially expressed proteins underscoring each proteome-based subtype, specific gene categories (by Gene Ontology, or GO, annotation) were over-represented (Fig. 3c), consistent with the correspondences noted above with the previous k1–k10 subtyping[21]. In addition, subtype s7 involved "axon guidance" and "frizzled binding" genes; subtype s10 involved "DNA repair" and "chromatin organization" genes; and subtype s11 involved "synapse," "dendrite," and "axon" genes. As applied to the proteomic data (Fig. 3d), gene signatures of immune cell types indicated the presence of T cells and higher expression of immune checkpoint pathway genes within s2

tumors. In contrast, s5 and s6 tumors showed signatures of B cells, eosinophils, and mast cells and higher expression of complement pathway genes. The above observations would reinforce the notion of different immune response pathways being active within different tumor subsets[21]. Within s4 and s11 tumors, proteomic signatures of metabolism indicated higher activation of pathways involving fatty acid metabolism, glycolysis and gluconeogenesis, pentose phosphate, TCA cycle, and oxidative phosphorylation, while s1, s6, and s8 tumors each showed higher signature levels of some but not all of the above pathways (Fig. 3d). On average, protein expression of matrix metalloproteinases (MMPs) was higher in s5, s6, and s8 tumors, while protein expression of collagen VI family was higher in s6 and s8 tumors (Fig. 3d). Canonical markers of neuroendocrine tumors, previously found over-expressed in ~4% of tumors in The Cancer Genome Atlas (TCGA) cohort[23], were higher on average in s6 and s11 tumors, the latter comprised of all brain tumors (Fig. 3d).

We examined proteomic datasets external to our proteomic compendium dataset for evidence of the manifestation of our proteome-based pan-cancer subtypes. Using a protein-based classifier developed from the proteomics compendium dataset, we classified 7694 TCGA tumors with Reverse-Phase Protein Array (RPPA) data according to s1–s11 proteome-based pan-cancer subtypes (Fig. 4a). The RPPA platform would be antibody-based and independent of the mass spectrometry platform. We had previously classified the TCGA RPPA profiles according to k1 through k10 subtypes[21], and the relationships between the s1–s11 and k1–k10 subtype classifications in the TCGA RPPA dataset mirrored the relationships observed in the proteomic compendium dataset (Figs. 4b and 3a). Also consistent with the proteomic compendium results, TCGA s3 tumors were highly enriched for basal-like breast cancer ($p < 1E{-}35$, one-sided Fisher's exact test), TCGA s11 tumors were highly enriched for brain tumors (TCGA GBM and LGG projects, $p < 1E{-}50$), and TCGA s10 tumor were moderately enriched for brain tumors ($p < 0.005$). Similarly, we classified 375 cancer cell lines with mass spectrometry-based proteomic data according to s1–s11 subtypes (Fig. 4c and Supplementary Data 4). Consistent with previous results[23], not all differential patterns observable in human tumor proteomic data appeared as strong in the cell line proteomic data, particularly regarding immune-associated or stroma-associated subtypes, attributable to various factors including growth conditions of cell lines lacking tumor microenvironmental effects. For 301 of the 375 cell lines, CRISPR knockout screens globally assessed gene essentiality[31]. In taking the global correlation between differential protein expression profile versus gene essentiality scoring profile for each cell line, s3 and s10 cell lines had consistent negative correlations in contrast to the other subtypes (Fig. 4d). This observation indicated that s3 and s10 cell lines (and, by extension, their tumor counterparts) tended to express essential genes highly (Fig. 4e).

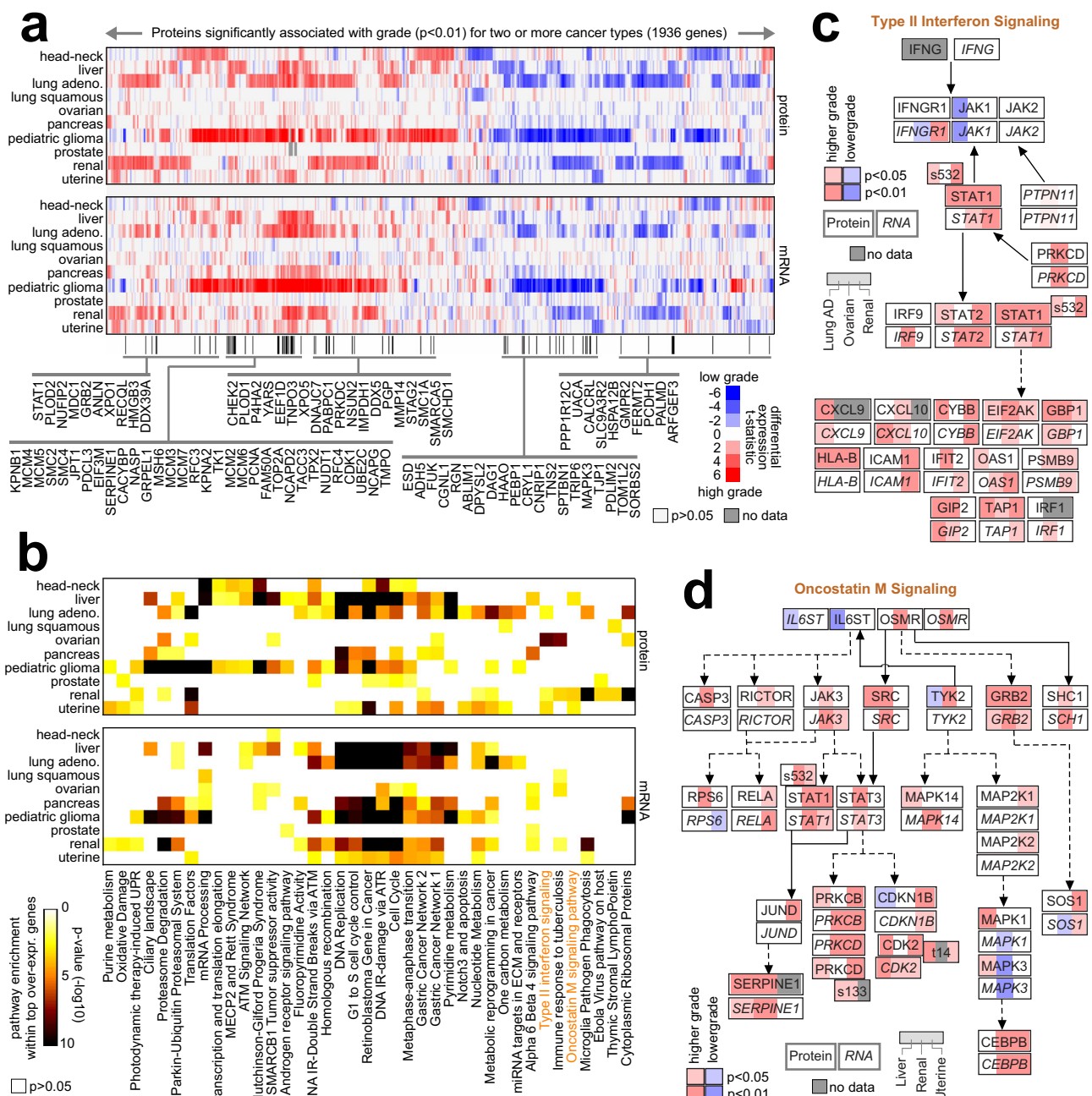

**Fig. 2 Proteomic and transcriptomic signatures of high-grade cancers. a** Across ten cancer types with tumor grade information, heat map of differential *t*-statistics (Pearson correlation on log-transformed data), by cancer type, comparing higher grade versus lower grade (red, higher expression with higher grade; white, not significant with *p* > 0.05), for 1936 proteins significant for two or more cancer types (*p* < 0.01). Differential *t*-statistics by grade for the mRNA corresponding to the 1936 proteins are also shown. Proteins significantly over-expressed (*p* < 0.01) with higher grade for four or more cancer types are indicated by name. **b** Significance of enrichment (by one-sided Fisher's exact test) for wikiPathway[28] gene sets with the respective sets of proteins and mRNAs over-expressed (*p* < 0.01, Pearson) with tumor grade for each cancer type represented. The pathways represented were significant within the over-expressed proteins for at least one cancer type with FDR < 10% and for at least two cancer types with *p* < 0.01. **c** Pathway diagram representing type II interferon signaling, with differential protein and mRNA expression patterns represented, correlating expression with increasing tumor stage for Lung AD (adenocarcinoma), Ovarian, and Renal cancer types. RNA features are indicated using italics. Phospho-protein features are indicated by residue. Red denotes significantly higher expression with higher grade, and blue denotes significantly lower expression. **d** Similar to **c**, but for Oncostatin M signaling and cancer types Liver, Renal, and Uterine.

The s10 and s11 subtypes (consisting of 350 and 441 tumors, respectively) involved brain tumors, with s10 enriched for pediatric brain tumors and s11 consisting entirely of brain tumors (pediatric and adult glioblastoma). These subtypes did not strongly correspond to any of the previous proteome-based subtypes[21], as the previous data did not involve brain tumors.

Proteins high in s11 tumors were highly enriched for brain tissue-specific genes by GTEX (Methods). By key somatic alteration events, protein and phospho-protein features, and pathway-associated proteomic signatures, s10 tumors showed increased alteration of the MYC, Wnt, and Hippo pathways compared to the rest of the tumors (Figs. 3d and 5a). Brain tumors of the

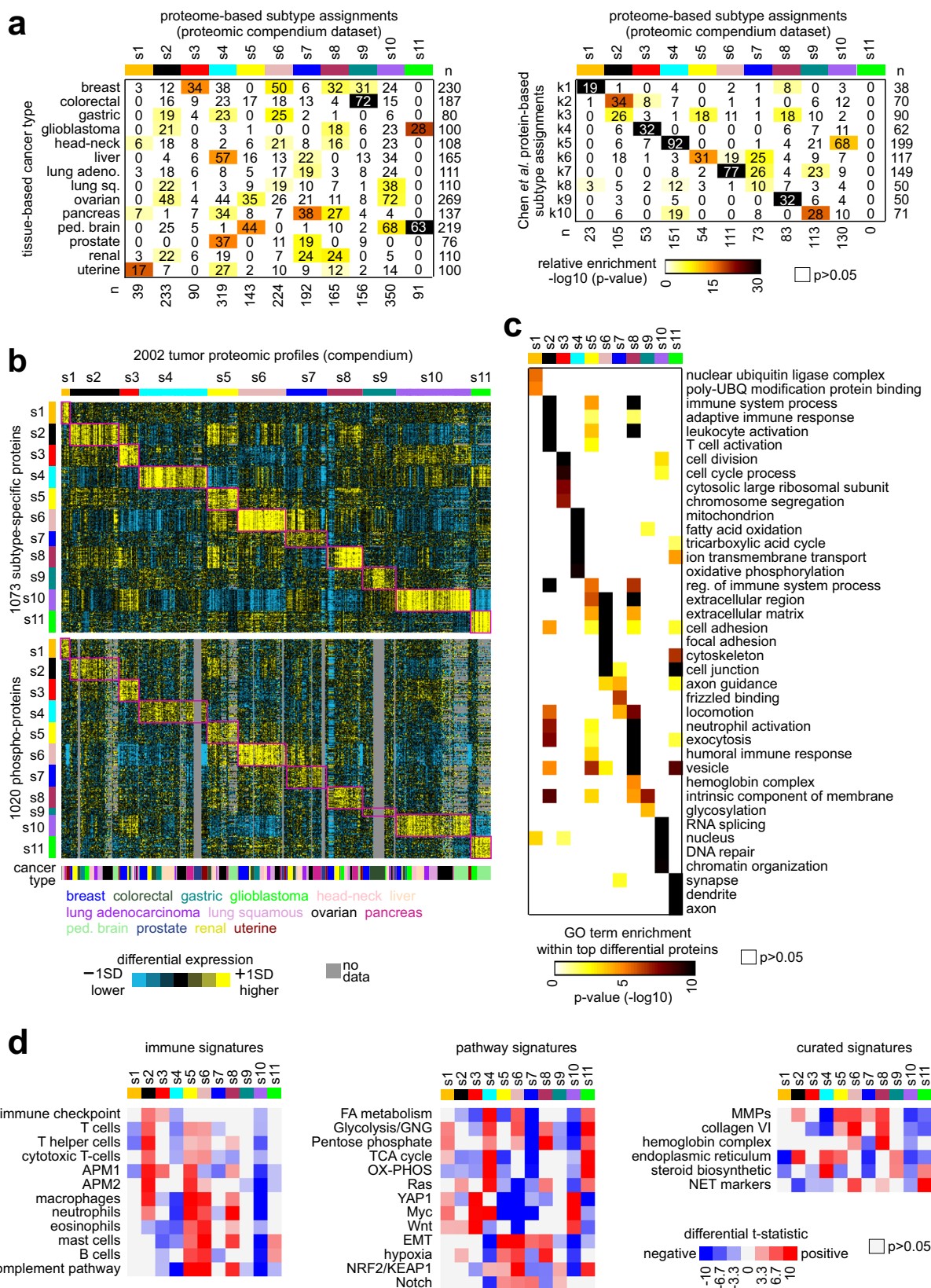

**a** proteome-based subtype assignments (proteomic compendium dataset)

**b** 2002 tumor proteomic profiles (compendium)

**c** GO term enrichment within top differential proteins

**d** immune signatures / pathway signatures / curated signatures

s10 subtype appeared molecularly distinct from s11 brain tumors. For example, s10 brain tumors showed lower expression of proteins involved in metabolic pathways, while s11 tumors showed higher average expression for these same pathways (Fig. 5b). In terms of previously identified glioblastoma subtypes

at the protein level[7], s10 glioblastomas associated with nmf3 classical tumors, while s11 glioblastomas associated with nmf1 proneural tumors (Fig. 5c). Concerning pediatric brain tumor histologic types, s10 tumors were associated with ependymomas and medulloblastomas, while s11 tumors were associated with

**Fig. 3 De novo pan-cancer proteomic subtypes. a** By ConsensusClusterPlus[55] of 2002 tumor proteomic profiles, ten proteomic-based subtypes—s1 through s11—were identified. Significance of overlap of these subtypes with tissue-based cancer types (left panel) and with previously identified pan-cancer subtypes[21] (k1–k10, right panel) is indicated. *p* values by one-sided Fisher's exact test. Lung adeno lung adenocarcinoma, ped. brain pediatric brain tumors. **b** Across 2002 tumor proteomic profiles, differential expression patterns (values normalized within each tissue-based cancer type; SD standard deviation from the median) for a set of 1073 proteins (top heat map) and for a set of 1020 phospho-proteins (bottom heat map) found to best distinguish between the ten proteome-based subtypes (see Methods, top ~100 over-expressed proteins for each subtype). **c** For the top over-expressed proteins associated with each subtype (from **b**, top panel), represented categories by GO were assessed, with selected enriched categories represented here (UBQ ubiquitin, reg regulation). *p* values by one-sided Fisher's exact test. **d** Proteomic signatures according to pathway or functional gene group (using values normalized within each cancer type). Red-blue heat maps denote *t*-statistics for comparing the given subtype versus the other tumors for each dataset. Selected pathways surveyed by signatures[23] included several related to metabolism (FA fatty acid, GNG gluconeogenesis, TCA tricarboxylic acid, OX-PHOS oxidative phosphorylation). APM1/APM2, antigen presentation on MHC class I/class II, respectively. NET neuroendocrine tumors.

---

gangliogliomas and low-grade gliomas (Fig. 5c). Our pediatric brain tumors were divided primarily among s2, s5, s10, and s11 subtypes (Fig. 3a). In pediatric brain tumors from the CBTTC cohort[14,32], the above subtypes involved differences in patient survival, both among the patients with tumors in the proteomic compendium and those with tumors having RNA-seq data only (Fig. 5d).

The proteome-based subtypes also showed differences from each other in overall CNA burden. The s10 and s3 subtypes had the highest CNA burden on average, and the s11 and s1 subtypes had the lowest CNA burden (Fig. 6a). A high rate of CNA within a given tumor may result in an altered molecular profile due to the extensive DNA damage involved[33]. Across all tumors, we computed the correlation between the expression of each gene and CNA burden at both protein and mRNA levels. We observed most of the significant gene-level associations at both the protein and mRNA levels. However, some genes were significantly correlated with CNA burden at the protein level but not at the mRNA level or vice versa (Fig. 6b). Genes increasing with higher CNA burden at both protein and mRNA levels involved chromatin, histone modification, transcription factor binding, cell cycle checkpoint, DNA repair, and methyltransferase complex (Fig. 6c). Genes increasing with higher CNA burden at the protein level but not at the mRNA level involved nuclear hormone receptor binding, histone deacetylation, and DNA repair. Genes increasing with higher CNA burden at the mRNA level but not at the protein level involved the electron transport chain, the NADH dehydrogenase complex, gluconeogenesis, and the nucleosome. Consistent with previous studies examining molecular correlates of overall structural variation burden across tumors[32,33], tumors with high CNA burden showed both high proteome-based signature scoring for DNA damage response pathways and low scoring for immune cell infiltrates, though with some genes in these signatures being significant by protein but not mRNA (Fig. 6d).

**Pathway-level somatic alterations**. We hypothesized that somatic alteration of well-characterized oncogenic or tumor suppressor pathways would be reflected in the cancer proteome, in terms of the downstream effects of altered pathway signaling. Across the entire proteomic compendium tumor dataset, assessment of genes within pathways demonstrated a high number of somatic alterations (small mutation or CNA) involving chromatin modification (73.4% of 1597 tumors involving 12 cancer types with small mutation and CNA data available), p53/Rb-related (73.1%), SWI/SNF complex (70.0%), PI3K/AKT/mTOR (69.2%), Receptor Tyrosine Kinase signaling (RTK, 55.0%), Wnt/beta-catenin (51.8%), MYC/MYCN (39.6%), NRF2 (8.2%), and Hippo signaling (36.3%) (Fig. 7a). The above pathways were altered in different ways involving different genes in different cancer types (Fig. S3). Expected associations of cancer type with somatic pathway alteration were observed, including *MYC* amplification

in breast cancer; Wnt pathway alterations via *APC* mutation in colorectal cancer; NRF2 pathway alteration in squamous lung cancer; *KRAS* mutations in colorectal, lung adenocarcinoma, and pancreatic cancers; *TP53* mutations in ovarian, lung squamous, and head and neck cancers; and mTOR pathway alterations via *VHL* mutation in renal cancer. As a means to integrate proteomic with somatic DNA alteration data, gene expression signatures for p53, k-ras, MTOR, Wnt/beta-catenin, MYC, Nrf2/Keap1, and YAP1/Hippo—based on data from experimental models—were applied to both the protein and the mRNA expression profiles of the tumors. We scored each tumor sample profile for each of the above pathways, with a higher signature score indicating a greater level of pathway activation in cells. For each pathway considered, relative levels of the corresponding signature—both at protein and mRNA levels—were significantly different between somatically altered versus unaltered tumors for that pathway, with the differences being in the anticipated direction (Fig. 3c). The one notable exception to the above was the p53 signature, where the mRNA-based but not the protein-based signature scoring was significantly associated with p53/Rb-related DNA alterations. The p53 signature consisted of 27 canonical p53 transcriptional targets[23] represented in the proteomic compendium dataset. At the mRNA level, 17 of the 27 p53 signature genes had a significant negative expression correlation (*p* < 0.05) with TP53 mutation status, while at the protein level, only 9 of the 27 genes had a similar negative correlation.

In addition to entire gene signatures, specific proteins could show differential levels in according to somatic pathway alteration. For example, we could define proteomic signatures associated with mutant *TP53* or mutant *KRAS* (Fig. 8a), representing two well-established cancer genes for which small mutation is the primary mode of pathway deregulation. Interestingly, a substantial number of proteins differentially expressed with *TP53* mutation were not reflected at the mRNA level (Fig. 8b). Out of 2752 proteins differential with *TP53* mutation (false discovery rate, or FDR < 10%), 1134 (41%) were not similarly altered significantly (*p* < 0.05) at the mRNA level (Supplementary Data 2 and 3). Similarly, out of 5414 genes with mRNA differential with *TP53* mutation (FDR < 10%), 2272 (42%) were not similarly altered at the protein level. *TP53* mutation associates with higher overall mutational burdens in cancer[32], and most of the proteomic signature of *TP53* mutation overlapped with the proteomic signature of CNA burden described above; at the same time, 638 (23%) of the 2752 *TP53* mutation-associated genes (FDR < 10%) were not significant in the same direction (*p* < 0.05) in the CNA burden analysis (Supplementary Data 2). The *KRAS* mutant gene signatures involved 199 proteins and 580 mRNAs (FDR < 10%), with 109 of the significant proteins showing the same trend (*p* < 0.05) at the mRNA level. Genes with expression higher with *TP53* mutation at both the protein and mRNA levels included genes involved in cell division, while genes higher specifically at the protein level involved SWI/SNF

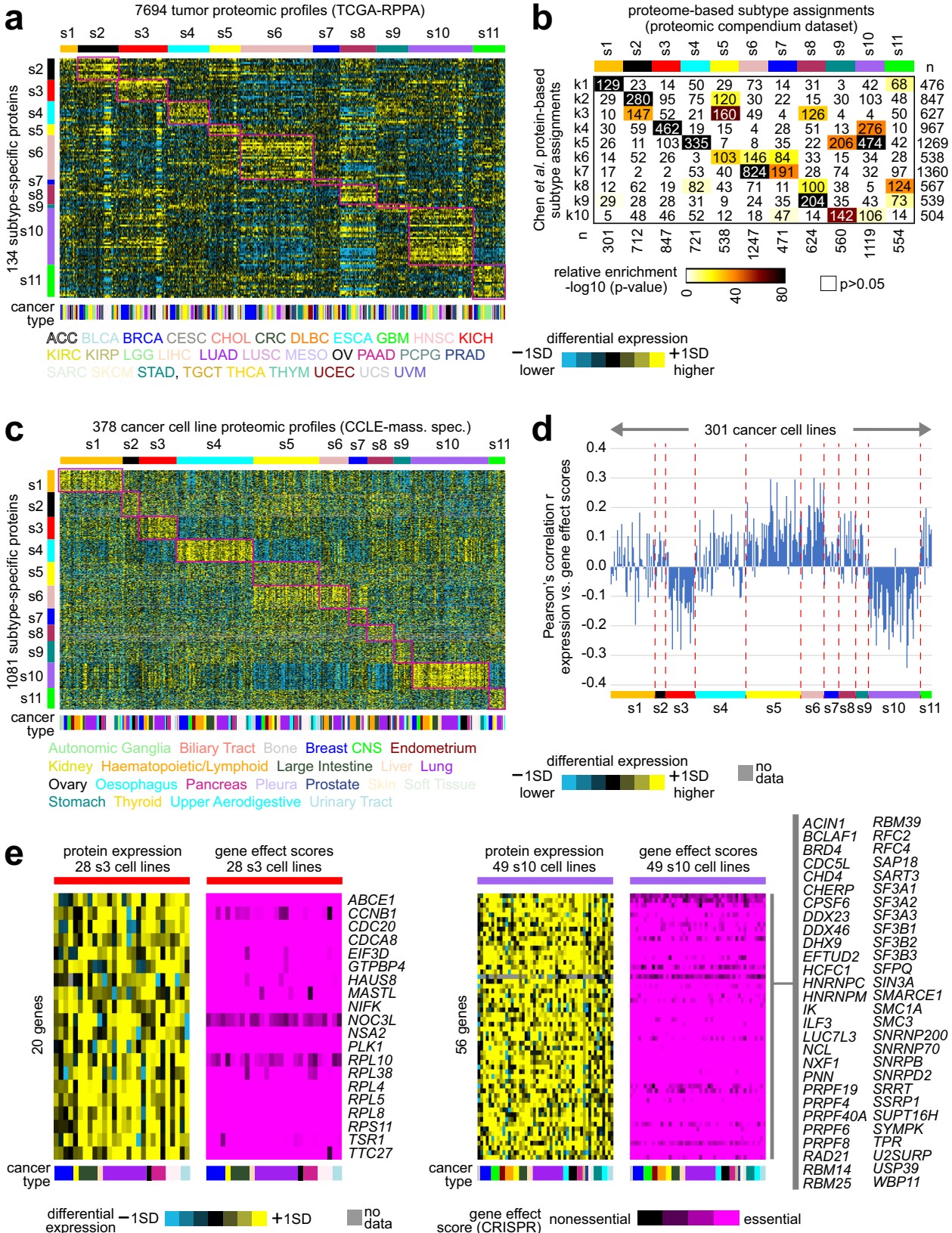

complex, and genes higher specifically at the mRNA level involved the proteasome, NF-kappaB signaling, and Wnt pathway (Fig. 8c). Genes with expression higher or lower with *KRAS* mutation at the mRNA level included genes involved in cell adhesion or humoral immune response, respectively (Fig. 8c).

Individual genes of interest associated with *TP53* or *KRAS* mutation at the protein but not mRNA levels would have known functional roles with the respective pathways (Supplementary Data 2 and 3). For example, protein levels for p53 and HEATR1 were higher with *TP53* mutation (Fig. 8d), where depletion of

**Fig. 4 Observation of pan-cancer proteome-based subtypes in independent proteomic datasets. a** The 7694 TCGA tumors with reverse-phase protein array (RPPA) data were classified according to proteome-based pan-cancer subtype (from Fig. 3b). Expression patterns for a top set of 134 proteins distinguishing between the 11 subtypes based on the proteomic compendium dataset (see the section "Methods", based on available data) are shown for the TCGA RPPA proteomic dataset. Gene patterns in the RPPA sample profiles sharing similarities with a subtype-specific signature pattern are highlighted. Cancer type represents TCGA project name. **b** For the TCGA dataset results, the significance of overlap of the s1–s11 proteomic subtypes with previously identified pan-cancer subtypes[21] (k1–k10) is indicated. *p* values by one-sided Fisher's exact test. **c** Cancer cell lines with mass spectrometry data in the Cancer Cell Line Encyclopedia (CCLE)[72] were classified according to proteome-based pan-cancer subtype (378 profiles representing 375 cell lines). Expression patterns for the top set of 1081 subtype-specific protein isoforms (based on the 1073 total proteins from Fig. 3b) are shown for the CCLE dataset. **d** For 301 of the 375 cell lines, based on the 1081 proteins in **c**, Pearson correlation between differential protein expression patterns versus gene effect scores based on Cancer Dependency Map (DepMap) CRISPR assays[31] (with low scores denoting essential genes), whereby s3 and s10 cell lines tend to have high expression of essential genes. **e** For s3 and s10 cell lines, associated patterns for the sets of genes having both high expression and low gene effect scores across cell lines (with at least half of cell lines having normalized expression >0.5 SD from the median and gene effect scores <−0.5). Cancer type color bar legend defined in **c**. SD standard deviation from the median.

HEATR1 would lead to impaired proliferation and induction of p53-dependent cell cycle arrest[34]. Proteins higher with *KRAS* mutation included INSR (insulin receptor)—where insulin can promote invasion and migration of *KRAS* mutant cells[35]—and ERBIN (erbb2 interacting protein)—which can regulate *KRAS* mutant-induced tumorigenesis[36].

Phospho-proteins are a component of pathway signaling, and, across the proteomic compendium, specific phospho-proteins of interest could be associated with pathway-level somatic alteration or pathway signature across tumors. Overall, phospho-proteins tended to be significantly positively correlated with the corresponding total protein, as expected (Supplementary Data 6). We considered a set of 106 phospho-proteins representing 25 genes involved in the pathway-level somatic alteration classes (from Fig. 7a), which phospho-protein features had sufficient data (>50% of tumors for seven or more cancer types). Many other phospho-proteins of interest, e.g., canonical pathway members of PI3K/AKT/mTOR[37], were not detected in enough tumor numbers across the proteomic compendium. On the other hand, 58 of the 106 phospho-proteins represented a phosphosite not cataloged in the Human Protein Reference Database (HPRD)[38]. Of the 106 phospho-proteins, 36 significantly correlated (*p* < 0.05, Pearson correlation) with the corresponding pathway-level DNA somatic mutation class (Fig. 9a, 15 of these representing uncatalogued phosphosites), and 24 significantly correlated with the corresponding pathway-level gene signature (Fig. 9b, 10 of these representing uncatalogued phosphosites), nine of which significantly correlated with pathway-level mutation class. Of the nine phospho-proteins, four, including PI3K/AKT/mTOR-related PTEN:s294 and RTK-related ERBB2:t671 and ERBB2:s968, were not found in HPRD (Fig. 9c).

A wide range of pathway gene signature levels was evident within somatically altered and unaltered groups for a given pathway. In addition to biological and technical noise that would be inherent in the data, there is the possibility that alterations in other genes may help drive a given pathway, even if that gene may not have a well-established or canonical role in that pathway[37]. In our proteomic compendium dataset, we examined the set of tumors with high pathway signature scoring but with no canonical mutations (see Methods). We aimed to identify any enrichment patterns for somatic small mutation events involving a set of 190 genes significantly mutated in pan-cancer studies[39,40] (Fig. 10a). We evaluated each of the MYC/MYCN, Hippo, mTOR, NRF2, p53/Rb-related, RTK, and Wnt/beta-catenin pathways in this way. We also carried out this analysis across 10,224 tumors in TCGA pan-cancer dataset (with signature scoring based on mRNA data). We looked for overlap in the respective results between the proteomic compendium and TCGA. In the proteomic compendium, 41 genes were significantly enriched for mutation events within pathway-activated

tumors with no canonical mutations (*p* < 0.01, one-sided Fisher's exact test), while 45 genes were significantly enriched for TCGA datasets (*p* < 0.001). Between the two tumor cohorts, we observed significant overlap between the respective gene-level associations (*p* < 5E−8, one-sided Fisher's exact test), involving 12 genes, all of which were enriched for mutation events in tumors with high MYC signature (Fig. 10b). For each dataset, somatic mutations in any of the 12 genes would account for a substantial number of tumors in addition to tumors with *MYC* or *MYCN* copy gain alone (Fig. 10c), on the order of an additional 14–18% of tumors surveyed. For each dataset, mutations in each of the 12 were associated with elevated MYC pathway signature scoring, like what we had observed above regarding *MYC* and *MYCN* copy gain or amplification (Fig. 10d). For several of the 12 genes, previous experimental studies have demonstrated a role for loss of the gene leading to increased MYC expression or activity, including *ARID1A*[41], *ATM*[42], *EP300*[43], *PTEN*[44], *RB1*[45], *SMARCA4*[46], and *ZFHX3*[47].

## Discussion

Across a large pan-cancer cohort of primary tumors, we observed general correspondence between protein and mRNA differential expression patterns, but with notable exceptions. Genes with proteins having less correlation across tumors with the corresponding mRNA or CNA patterns included genes involved in ribosomes, translation, electron transport chain, humoral immune response, DNA repair, or cell cycle regulation. Genes with expression increasing with higher CNA burden at only the protein level included genes involved in nuclear hormone receptor binding, histone modification, and DNA repair. Pathways associated with high-grade cancers at the protein but not the mRNA level included type II interferon and Oncostatin M signaling. A gene signature of p53 transcriptional targets applied to mRNA data showed an expected association with p53 pathway-associated mutations, while the signature applied to protein data did not show a similar result. Widespread differential protein expression patterns could be associated with somatic mutation of *TP53* or *KRAS*, with some patterns not reflected at the mRNA level. A lack of correlation or association represents a negative result, inherently difficult to prove using statistics, and technical issues regarding the data platforms might conceivably be at play. At the same time, the fact that the sets of genes involved in the overall differences observed between protein and mRNA are significantly enriched for specific functional groups and annotated pathways would suggest that true biology is involved here. Our study reinforces the notion that cancers should be comprehensively surveyed at the protein level, where expression profiling on tumors has historically been mostly limited to the transcript level.

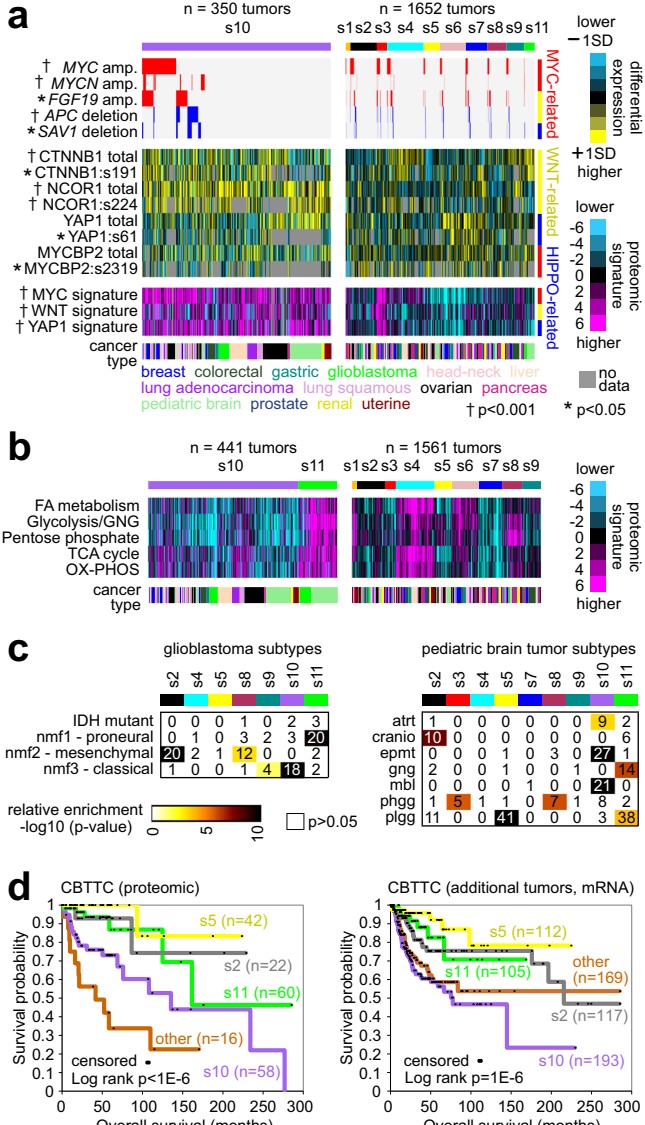

**Fig. 5 Pan-cancer proteomic subtypes involving brain tumors. a** Across s10 tumors, as compared to other subtypes, selected molecular features involving MYC, Wnt/beta-catenin, and Hippo pathways. Enrichment *p* values by one-sided Fisher's exact test for amplification ("amp.") and deletion features. *p* values for higher expression of protein, phospho-protein, and proteomic signatures by *t*-test. SD standard deviation from the median. **b** Across s10 and s11 tumors, as compared to other subtypes, selected proteomic signatures involving metabolic pathways. **c** For the proteomic-based pan-cancer subtypes involving glioblastomas (left panel) and pediatric brain tumors (right panel), the respective significances of overlap with previously identified glioblastoma subtypes[7] and with pediatric brain tumor histologic types (atrt atypical teratoid rhabdoid tumor, cranio craniopharyngioma, epmt ependymoma, gng ganglioglioma, mbl medulloblastoma, phgg high-grade glioma/ astrocytoma, plgg low-grade glioma/astrocytoma). *p* values by one-sided Fisher's exact test. **d** In the CBTTC pediatric brain tumor datasets, differences in patient overall survival among the pediatric brain-associated proteomic-based subtypes. Left panel represents tumors in CBTTC with proteomic data. Right panel represents the remaining CBTTC tumors with RNA-seq data but not proteomic data, whereby RNA-seq profiles were assigned a proteomic subtype. *p* values by log-rank test. For the RNA-seq results, differences remain significant (*p* = 0.002) after correcting for histologic type (by stratified log-rank test).

The proteome-based molecular subtypes identified using our tumor cohort reinforce the notion of pan-cancer molecular classes cutting across tumor lineages and cancer types[21,23]. Most of the proteome-based subtypes identified previously in a more limited cohort[21] were re-discovered here by unsupervised analyses of a larger cohort with additional cancer types. We again observed multiple distinct subtypes involving the immune system, one involving the adaptive immune response and T-cell activation, and others associated with the humoral immune response and complement pathway. We also again observed tumor stroma-associated subtypes, one involving collagen VI network. This present study extends the previously reported subtype-related findings to additional cancer types. Importantly, we could also identify additional proteome-based subtypes, including one enriched for brain tumors and associated with MYC, Wnt, and Hippo pathways and with high CNA burden, and another consisting entirely of brain tumors and associated with increased expression of genes in metabolic pathways and with low CNA burden. As shown here, proteome-based subtyping can provide insights into the pathways and processes appearing deregulated within tumor subsets, suggesting therapeutic opportunities. Differences in subtyping assignments between the present study and previous studies could involve differences in the respective cohorts, including represented cancer types. As more human tumors and additional cancer types are profiled by mass spectrometry-based proteomics, additional subtypes may be uncovered and explored in future studies.

We found that somatic alterations of cancer-associated pathways are reflected in the cancer proteome, whereby tumors with somatic alteration involving genes in a pathway tend to show higher levels of protein-based signature scoring for that pathway. Our study took advantage of the large number of tumors represented in our proteomic compendium dataset, whereby we could carry out tumor subset analyses involving sparse mutation events. As cataloged previously, the gene signatures represent the downstream effects of altered pathway signaling, e.g., the upregulation of transcriptional targets in a cell line or mouse model. The proteomic and genomic data integration represents orthogonal information pointing to a common tumor subset deregulated for a given pathway. Our collective knowledge of molecular pathways has been largely derived from experimental models. The signature analyses allow us to explore experimentally inferred cause-and-effect relationships in the human disease setting, whereby these relationships manifest in tumors as significant correlations. Phospho-proteins can denote pathway signaling, though current mass spectrometry-based datasets may not capture all phospho-protein features of interest regarding a pathway. However, post-translational modifications, including phosphorylation events, may be detected by mass spectrometry. Pathway activity, as measured using gene signatures applied to proteome data, reflects known mutations or copy alteration in most but not all tumors examined, suggesting additional, unexplained, or underappreciated mechanisms of pathway activation. In the case of the MYC pathway, we identified several genes, many of which would have an underappreciated role in the MYC pathway, for which somatic mutation involved higher pathway activation. Through proteogenomics, additional members or connections may be incorporated into the standard pathway model.

Our study results provide a framework for understanding the molecular landscape of cancers at the proteome level. The associated datasets and gene-level associations represent a resource for the research community, including helping to identify gene candidates for functional studies. Beyond the additional tumors and cancer types represented in our proteomic compendium of 2002 tumors, the pan-cancer datasets compiled here involve multiple levels of molecular data in addition to the proteome,

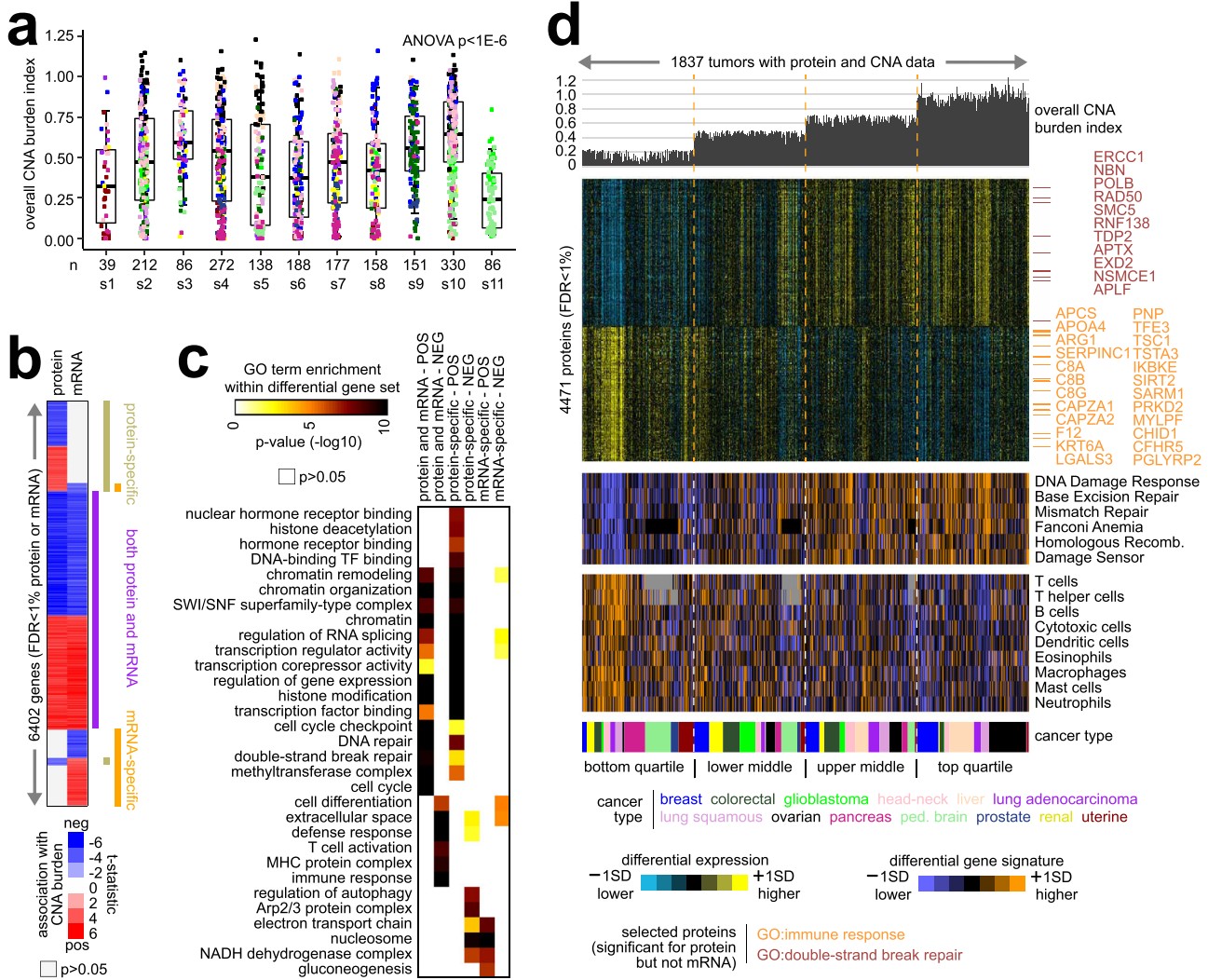

**Fig. 6 Global proteomic alterations associated with the overall CNA burden across cancers. a** Overall CNA burden index (standard deviation of CNA values across all genes) by proteome-based molecular subtype. Box plots represent 5% (lower whisker), 25% (lower box), 50% (median), 75% (upper box), and 95% (upper whisker). n = biologically independent tumors. p value by ANOVA. Data points are colored according to cancer type (legend in **d**). **b** Protein and mRNA features correlated with CNA burden index (FDR < 1%) across all cancers studied (n = 1837 tumors with protein and CNA data; n = 1748 with RNA and CNA data), including genes with associations by protein but not mRNA ("protein-specific"), genes with associations by mRNA but not protein ("mRNA-specific"), and genes with associations by both protein and mRNA ("both protein and mRNA"). **c** For the protein and mRNA signature from **b**, represented categories by GO were assessed, with selected enriched categories represented here. p values by one-sided Fisher's exact test. **d** Across the 1837 tumors with protein and CNA data, with tumors ranked high to low by global CNA index quartiles, selected molecular features are represented, including top protein correlates with CNA burden, proteome-based signatures scoring for DNA damage response pathways[52], and proteome-based scoring for immune cell infiltrates[67]. Proteins highlighted by name have GO annotation "double-strand break repair" or "immune response" and were significant for protein but not mRNA.

including mRNA-level and DNA-level data on the same set of tumors. In contrast, due in part to the timing of data releases, our first pan-cancer proteomic tumor subtyping study focused mainly on proteomic data in CPTAC tumors[21], while our recent study examining tumor grade correlates incorporated both protein and mRNA data but no somatic DNA-level data on CPTAC tumors[22]. Our present proteogenomics survey enabled us to explore the interplay between genome and proteome, including aspects not represented in the transcriptome. When previously surveying proteomic correlates of tumor grade, we identified specific protein kinases having functional impact in vitro in uterine endometrial cancer cells, which provides a template for other researchers to utilize the gene-level associations provided in the present study. In addition to correlates of tumor grade, proteomic correlates of somatic alteration of pathways, including *TP53* or *KRAS*

mutation, may be of interest for further study. Regarding the pan-cancer proteome-based subtypes, a number of these appear manifested in cancer cell lines, and additional data, such as DepMap gene dependency data, may be leveraged to help select targets of interest to examine in these cell lines. RPPA profiling data can classify tumors according to subtype, where RPPA antibody-based features may also lend themselves to relevant immunohistochemistry studies. For all cancer types studied, we have added the proteomic datasets to the user-friendly UALCAN data portal[26,48], facilitating differential analyses by protein, giving the research community ready access to our results.

## Methods

**Proteomic datasets.** We assembled a compendium dataset of mass spectrometry-based proteomics data of primary tumors from 14 cancer types and 17 individual

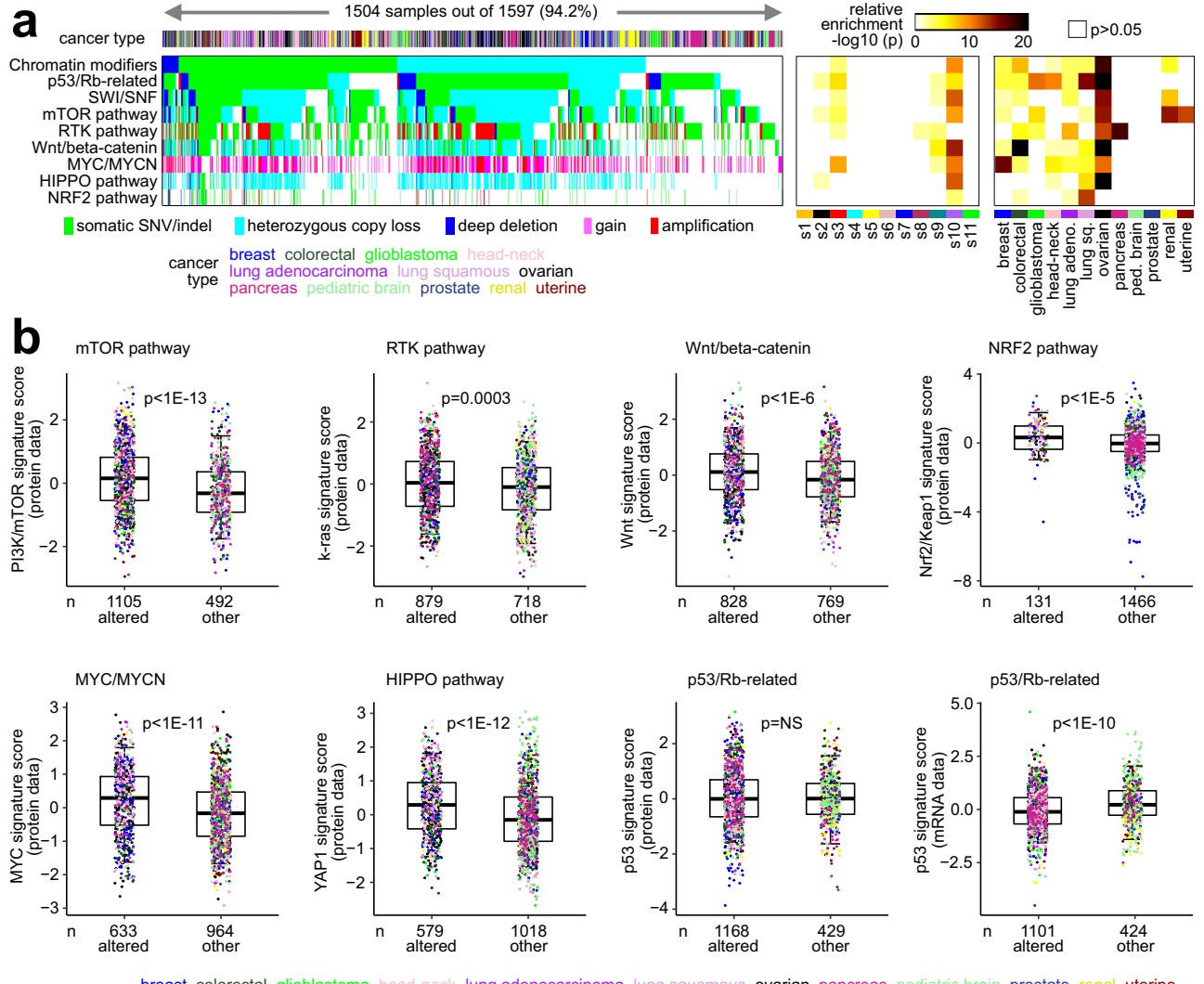

**Fig. 7 Somatic mutations and associated pathways.** **a** Pathway-centric view of somatic alterations across 1597 tumors (1504 with alteration of at least one pathway) with combined proteomic, whole-exome, and CNA data, involving key pathways and genes previously annotated across multiple cancer types based on domain knowledge (see Methods)[23,25,57,68]. Panels on the right represent the significance of enrichment (one-sided Fisher's exact test) of gene alteration events for each pathway within any proteome-based molecular subtype or tissue-based cancer type versus the rest of the tumors. RTK Receptor Tyrosine Kinase. **b** Pathway-associated gene signatures were applied to the proteomic profiles, while a p53 gene signature was applied to both proteomic and transcriptomic profiles. Box plots compare pathway-altered versus -unaltered tumors for relative levels of the corresponding signature. *p* values by *t*-test. Box plots represent 5% (lower whisker), 25% (lower box), 50% (median), 75% (upper box), and 95% (upper whisker). *n* = biologically independent tumors.

studies[3–19] (Supplementary Data 1). Most of these previous studies were led by either CPTAC or the International Cancer Proteogenome Consortium (ICPC). The cancer types represented in the proteomics compendium dataset were the following: Breast Invasive Carcinoma ($n = 230$ tumors with proteomics data)[3,4], Colorectal Adenocarcinoma ($n = 187$)[5,6], Gastric Cancer ($n = 80$)[18], Glioblastoma ($n = 100$)[7], Head and Neck Squamous Cell Carcinoma ($n = 108$)[8], Hepatocellular Carcinoma ($n = 165$)[17], Lung Adenocarcinoma ($n = 111$)[9], Lung Squamous Cell Carcinoma ($n = 110$)[10], Ovarian Serous Cystadenocarcinoma ($n = 269$)[11,12], Pancreatic Ductal Adenocarcinoma ($n = 137$)[13], Pediatric Brain Tumors ($n = 219$)[14], Prostate Adenocarcinoma ($n = 76$)[19], Renal Cell Carcinoma ($n = 110$)[15], and Uterine Corpus Endometrial Carcinoma ($n = 100$)[16]. The 2002 tumors in the compendium represented 1982 patients, with the pediatric brain tumor dataset involving 219 tumors from 199 patients. The above studies analyzed the tumors using global proteomic and phosphoproteomic profiling by liquid chromatography-tandem mass spectrometry (LC-MS/MS). We obtained processed protein expression data from the CPTAC Data Portal[49] or the associated publications' supplementary tables. Proteomic data, as provided by the CPTAC Data Portal and related publications, were processed at the gene level rather than at the protein isoform level; as a simplification, we did not consider different isoforms for the same protein in the present study.

For each study, taking the expression values provided in the associated data table, we normalized proteomic data for downstream analyses in the following

manner and as previously described[21,22]. First, within each proteomic profile, we normalized logged expression values to standard deviations from the median. Next, we normalized expression values across samples to standard deviations from the median. Similarly, we separately normalized both total protein and phospho-protein datasets for a given cancer type and dataset. For datasets where two different data centers generated values on the same tumors, we averaged normalized values from the respective data centers in instances of duplicate profiles for the same tumor sample. As intended, by normalizing expression within each cancer type and within each proteomic dataset, neither tissue-dominant differences nor inter-laboratory batch effects would drive the downstream analysis results, including unsupervised subtype discovery. For the compendium dataset of total proteins, a total of 15,439 unique genes by Entrez Identifier were represented in at least one of the individual datasets. For the compendium dataset of phospho-proteins, a total of 199,284 phospho-proteins, involving 11,671 unique genes, were represented in at least one of the individual datasets. Of these phospho-proteins, 5419 had available data for >50% of samples in at least seven cancer types.

**Transcriptomic datasets.** Of the 2002 human tumors with proteomics data, 1899 had corresponding RNA-seq data. For CPTAC projects utilizing tumors from TCGA, we obtained TCGA data RNA-seq data from the Broad Institute's Firehose

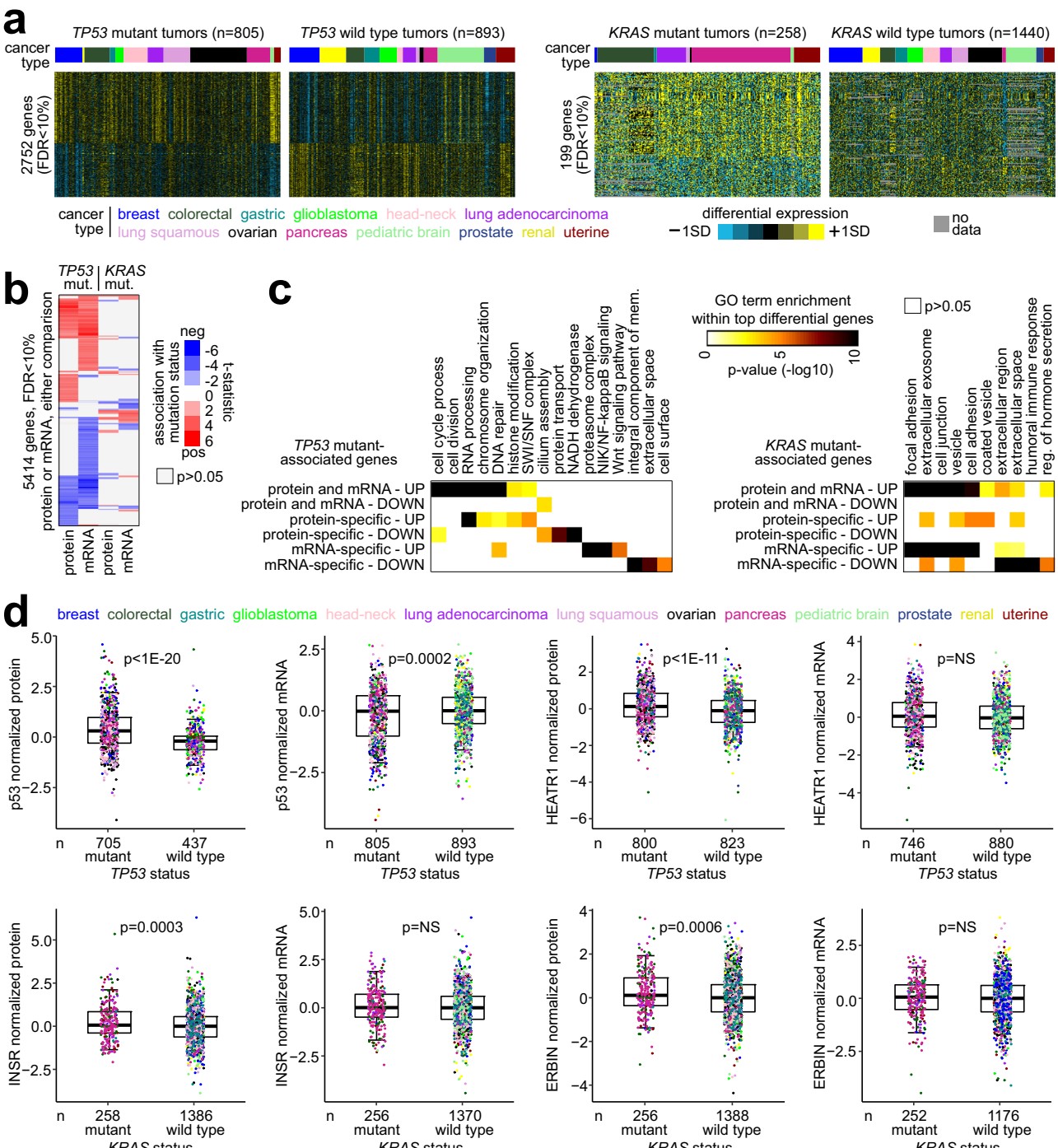

**Fig. 8 Proteomic and transcriptomic signatures associated with somatic mutation of *TP53* or *KRAS*. a** Differential expression heat maps of top total protein features correlated with small somatic mutation of *TP53* (left, FDR < 10%) or with somatic hotspot mutation of *KRAS* (right, FDR < 10%) across all cancers studied (*n* = 1440 tumors with protein and mutation data). SD standard deviation from the median. **b** Across *TP53* and *KRAS* comparisons, heat map representation of differential statistics for protein and mRNA features correlated with mutation status (FDR < 10%). **c** Taking the genes from **b**, gene sets were defined according to mutation association by protein versus mRNA (using *p* < 0.05 with the 5414 genes significant with FDR < 10% for any comparison). For each mutation signature analysis, genes significant (up in mutant versus down) for both protein and mRNA or for one molecular level but not the other were assessed for represented categories by GO, with selected enriched categories represented here. *p* values by one-sided Fisher's exact test. **d** For selected genes (*TP53*, *HEATR1*, *INSR*, *ERBIN*), box plots compare *TP53* or *KRAS* mutant versus wild-type tumors for relative levels of protein versus mRNA, as indicated. *p* values by *t*-test. Box plots represent 5% (lower whisker), 25% (lower box), 50% (median), 75% (upper box), and 95% (upper whisker). *n* = biologically independent tumors.

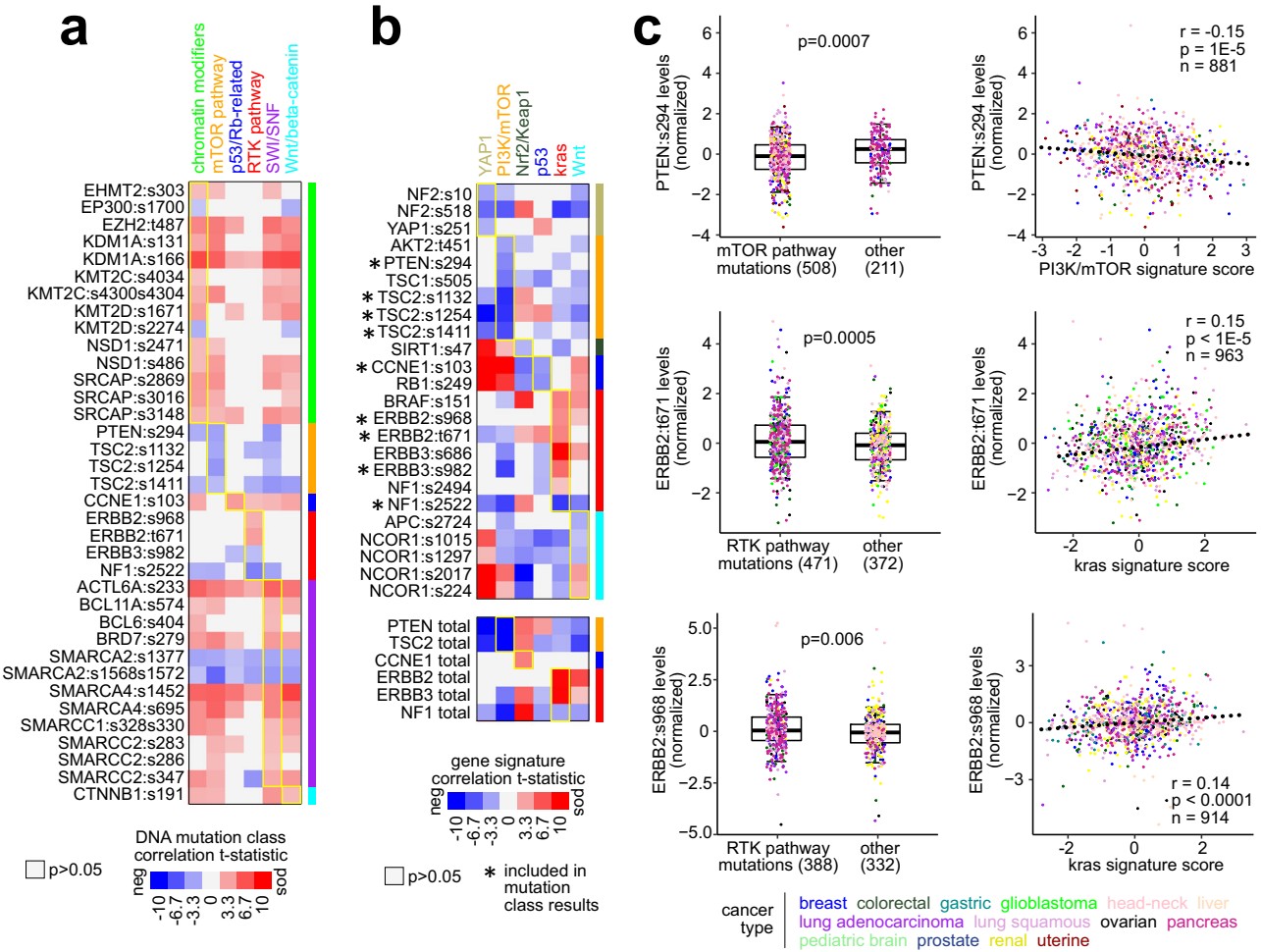

**Fig. 9 Phospho-proteins associated with pathway alteration. a** For 106 phospho-proteins (with data for >50% of samples for seven or more cancer types) with genes involved in curated pathway sets[23,25,57,68], the Pearson correlation with each pathway-level DNA mutation class from Fig. 7a was computed. Shown are 36 of the 106 phospho-proteins for which a significant association (*p* < 0.05) with its corresponding pathway mutation class was found. **b** For the 106 phospho-proteins, the Pearson correlation with each pathway gene signature from Fig. 7b was computed. Shown are 24 phospho-proteins for which a significant association (*p* < 0.05) with its corresponding pathway signature was found. Nine phospho-proteins are shared between **a** and **b** results. **c** For selected phospho-proteins PTEN:s294, ERBB2:t671, and ERBB2:s968, associations with mTOR pathway or RTK pathway mutation class (left) and association with mTOR pathway or RTK pathway signature (right) is represented. *p* values by *t*-test (box plots) or Pearson correlation (scatter plots). Box plots represent 5% (lower whisker), 25% (lower box), 50% (median), 75% (upper box), and 95% (upper whisker). *n* = biologically independent tumors.

data portal [https://gdac.broadinstitute.org], and we obtained RNA-seq data for the other CPTAC projects from the Genome Data Commons [https://gdc.cancer.gov/]. We obtained RNA-seq data on CBTN pediatric brain tumors through the public project on the Kids First Data Resource Portal and Cavatica [https://cbtn.org]. We obtained data for the non-CPTAC projects from links or accession numbers provided with the associated publications. We normalized expression values across samples to standard deviations from the median within each cancer type and dataset, as we carried out above for proteomic data.

**Copy number alteration (CNA) datasets**. Of the 2002 human tumors with proteomics data, 1837 had corresponding gene-level CNA data. No gene-level CNA data were available for the ICPC Gastic Cancer project (as these were not a part of the original study). For CPTAC projects utilizing tumors from TCGA, we obtained SNP array-based CNA "thresholded" values (−2, −1, 0, 1, 2) from the Broad Institute's Firehose data portal [https://gdac.broadinstitute.org]. We obtained gene-level absolute copy data (0, 1, 2, 3, 4, 5) for the other CPTAC projects from the Genome Data Commons [https://gdc.cancer.gov/]. We first normalized the absolute copy data according to ploidy (dividing gene copy value by average copy value for all genes), then thresholded to values approximating homozygous deletion (−2), heterozygous deletion (−1), wild-type (0), gain of 1–2 copies (+1), and amplification with at least 5 copies (2). We obtained gene-level copy data from CBTN pediatric brain tumors from Cavatica [https://cbtn.org] and thresholded similarly to the CPTAC copy data. For the prostate cancer dataset, gene-level copy

data based on WGS data was previously generated by the Pan-cancer Analysis of Whole Genomes consortium from a consensus of multiple CNA callers[50]. For the hepatocellular carcinoma project, we obtained whole-exome-based gene-level thresholded CNA calls from the National Omics Data Encyclopedia [https://www.biosino.org/node/project/detail/OEP000321].

We computed an overall CNA burden index for each CNA profile, defined as the standard deviation of thresholded CNA values across all genes. The CNA burden index was high in tumors with many copy gain/amplification or loss/deletion events. Across all samples, we computed the Pearson correlation of each protein or mRNA feature (using normalized values) with the CNA burden index. The method of Storey and Tibshirani[51] estimated FDR for genes significantly associated with the overall CNA burden. Using an alternative regression model incorporating cancer type did not improve the overall results, as these did not depend on cancer type differences (as we normalized all gene expression features within each cancer type, removing cancer type-specific differences). In considering proteins versus mRNA features significantly associated with CNA burden, we considered the 10,315 genes for which there were both protein and mRNA data for more than 500 tumors. For gene signatures of DNA damage response pathways, we obtained curated gene sets from ref. [52] and took the average of the normalized gene values.

**Small mutation datasets**. Of the 2002 human tumors with proteomics data, 1698 had corresponding small somatic mutation data (SNVs and indels) by whole-exome or whole-genome sequencing. No small mutation data were available for the ICPC

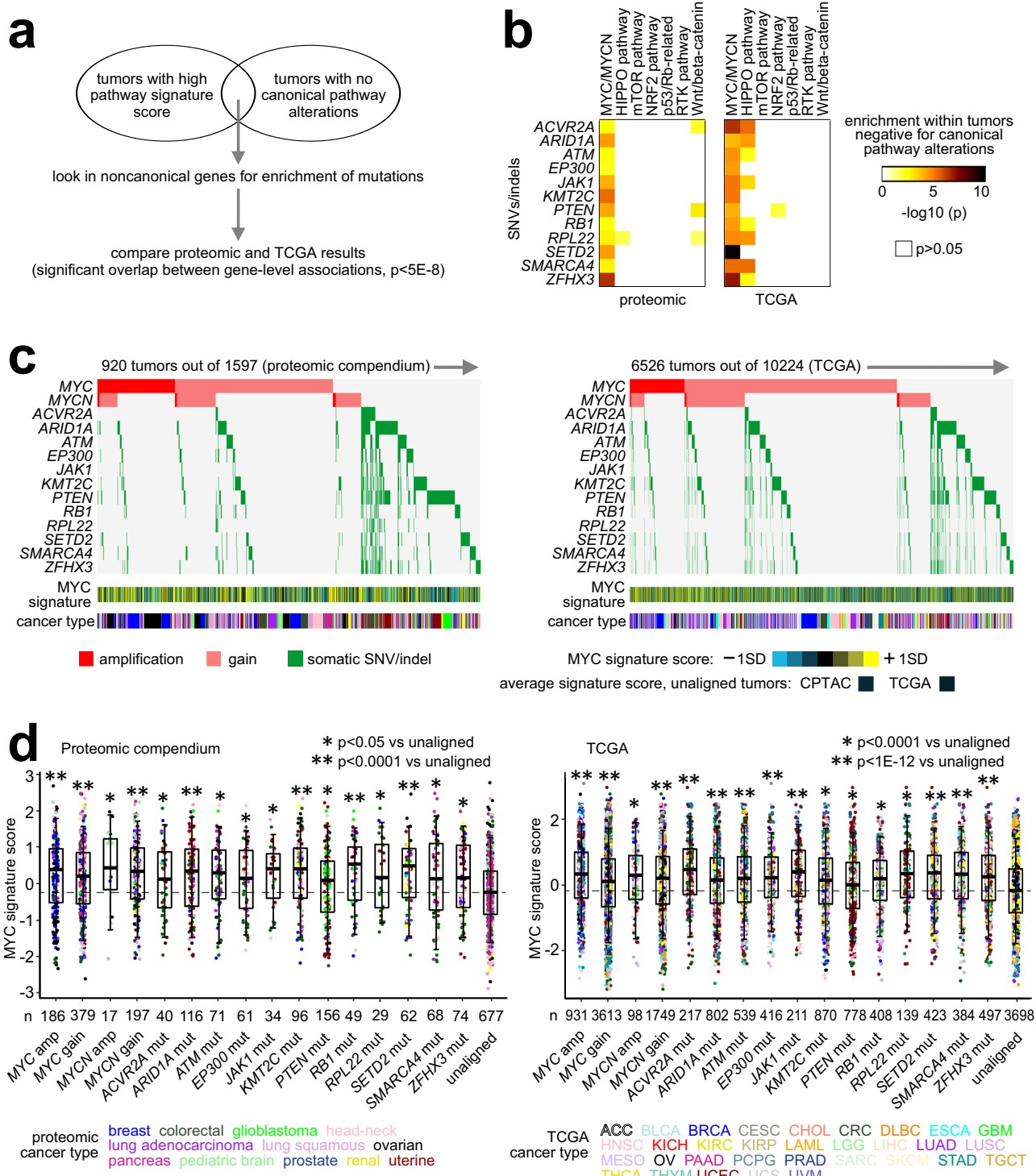

**Fig. 10 Noncanonical somatic mutation associations with MYC family alterations. a** Schematic of approach to identify somatic mutations (SNVs or indels) associated with a given pathway. Taking the set of tumors with high pathway signature scoring but with no canonical mutations (based on Fig. 7), we consider noncanonical genes for enrichment of mutations. Results are compared between the proteomic compendium and The Cancer Genome Atlas (TCGA), where we observed significant overlap between the respective gene-level associations (*p* < 5E−8, one-sided Fisher's exact test). **b** List of top mutated genes overlapping between proteomic compendium and TCGA results (*p* < 0.01 and *p* < 0.001, respectively, one-sided Fisher's exact test). Results involve 12 genes with mutation events enriched within tumors with high MYC signature but no *MYC* or *MYCN* copy gain or amplification. **c** Across both the proteomic compendium (left) and TCGA (right) tumor datasets, heat maps of *MYC/MYCN* copy alteration and of somatic mutation (SNV or indel) in genes from **b**. SD standard deviation from the median. Cancer type color bar legend defined in **d**. **d** Box plots of MYC/MYCN signature scores by alteration class in proteomic compendium (left) and TCGA (right) tumor datasets. *p* values versus unaligned group (tumors with none of the listed alterations) by *t*-test. Box plots represent 5% (lower whisker), 25% (lower box), 50% (median), 75% (upper box), and 95% (upper whisker). *n* = biologically independent tumors. Cancer type represents TCGA project name.

Liver Cancer project. For CPTAC projects utilizing tumors from TCGA, we obtained somatic mutation calls by whole-exome sequencing from the publicly available "MC3" TCGA MAF file [https://www.synapse.org/#!Synapse:syn7214402]; variants called by two or more algorithms were used in this study. For the other CPTAC projects, we obtained whole-exome somatic mutation calls from the Genome Data Commons [https://gdc.cancer.gov/]; variants called by two or more algorithms were used in this study. For the prostate cancer dataset, we obtained whole-genome somatic mutation calls from the supplemental of ref. [53]. Somatic SNV calls from the ICPC Gastic Cancer project were kindly provided by the authors of the associated study[18].

In considering proteins versus mRNA features significantly associated with small mutation status of TP53 or KRAS, we considered the 8734 genes for which there were both data for more than 1000 tumors and for >50% of samples in at least seven cancer types. For TP53, any somatic SNV or indel were considered. For KRAS, somatic SNVs occurred in "hotspot" residues reported by ref. [54] were considered. Differential expression between comparison groups was determined using t-test on the expression levels as normalized within each study, with FDR estimated using the method of Storey and Tibshirani[51].

**Pan-cancer molecular subtype discovery**. ConsensusClusterPlus R-package[55] (using R version 4.1.1) was used to identify the structure and relationship of the samples. For unsupervised clustering analysis, we selected the top 2000 most variable proteins from the proteomic compendium dataset of total protein (taken from the set of 3976 unique proteins represented in at least half of samples for all 14 cancer types represented), according to average standard deviation (using log-transformed expression values centered to standard deviations from the median within each cancer type) across the 14 cancer types. Consensus ward linkage hierarchical clustering identified $k = 2$ to $k = 20$ subtypes, with the stability of the clustering increasing with increasing k. We considered multiple subtype solutions, as described in Fig. S2. Beyond $k = 13$, we observed no further meaningful separation of the tumors as representative of distinctive biology. When further examining the 13-subtype solution, some subtypes appeared similar to each other on a visual inspection of the differential expression patterns. Therefore, we consolidated these subtypes into one, which resulted in a final 11 subtype solution. Our previous proteome-based molecular subtyping study[21] had referred to its set of subtypes as k1 through k10, based on the "k" in the k-means algorithm used in ConsensusClusterPlus. In this present study, we referred to the second set of subtypes of s1 through s11—the "s" representing "subtype"—as we wanted to draw distinctions between the respective sets of cluster assignments.

We examined an RNA-seq dataset of pediatric brain tumors from CBTTC, classifying each tumor RNA-seq profile by pan-cancer proteome-based subtype. Of the 909 unique patients represented in the CBTTC dataset, 696 did not have proteomic data. Where multiple tumors were taken from the same patient, we considered only one tumor profile in the downstream survival analyses. For the RNA-seq profiles being classified, we first normalized log2-transformed genes or proteins to standard deviations from the median. As a classifier, we used the top set of 1073 proteins (from the present study, Fig. 3) distinguishing between the pan-cancer subtypes. For each pan-cancer subtype, we computed the average normalized value for each protein, based on the centered expression data matrix. We then computed the Pearson correlation between each CBTTC RNA-seq profile and each pan-cancer subtype averaged profile. We assigned each RNA-seq profile to a pan-cancer subtype, based on which subtype profile showed the highest correlation with the given external dataset profile.

**Differential expression analyses**. We assessed differential expression between comparison groups using t-tests on expression values log2 transformed and normalized within cancer type as described above. For the top pan-cancer proteome-based subtype correlates (both for total protein and phospho-protein), normalized expression levels of each subtype were compared with the rest of the tumors by t-test. For defining the top over-expressed protein features for each subtype were first compared with the rest of the tumors. For a given subtype, a top protein had the highest differential expression by t-statistic compared to that of the other subtypes and a higher t-statistic compared to the other proteins that did not make the top list. When defining the top 100 over-expressed proteins for each subtype, in some cases, a subtype may have fewer than 100 top proteins (e.g., for the s7 subtype, where more under-expressed total proteins versus over-expressed proteins were associated with the subtype). For the differential analysis of phospho-proteins, we focused on the 5419 phospho-proteins with available data for >50% of samples in at least seven cancer types.

To determine differential levels for a given protein or mRNA according to increasing tumor grade, we took the Pearson correlation between the log-transformed gene-level molecular values and the grade translated into a numerical value. When converting tumor grade into a numerical variable, grade categories such as "G1", "G2", and "G3", for example, were translated as 1, 2, and 3, respectively, for differential analyses. For each cancer type, we selected for downstream analyses the set of differential proteins or mRNAs with Pearson p value <0.01. Even in instances of nominally significant proteins that moderately exceed chance expectations by estimated FDR, the top proteins may still contain molecular information representing real biological differences. For example, most

of the top protein lists examined showed significant enrichment patterns by wikiPathways analysis. For the pediatric brain tumor dataset, grade comparisons were restricted to pediatric gliomas (comparing high-grade versus low-grade tumors).

We evaluated enrichment of GO annotation terms[27] and wikiPathways[28] within sets of differentially expressed genes using SigTerms software[56] and one-sided Fisher's exact tests, with FDRs estimated using the method of Storey and Tibshirini[51]. GO annotation gene sets with less than ten genes were not considered for top results by FDR. Gene sets for each wikiPathway were downloaded in July 2019 ("20190710" version). For GO term enrichment analysis, we used all 10129 unique proteins represented in half of the tumors for at least seven cancer types profiled as the reference population. We searched each proteomic signature of the total proteins for enriched GO terms, with "high" proteins evaluated separately from "low" proteins. For wikiPathways enrichment analysis, we used all 6597 unique proteins represented in at least one wikiPathway as the reference population.

Proteins high in s11 tumors were highly enriched for brain tissue-specific genes by GTEX [https://www.gtexportal.org]. Taking the GTEX dataset, the 11,311 genes with average TPM values greater than 5 and with Entrez ID represented in our proteomic compendium dataset were considered. From GTEX, we selected the top 10% of genes with the most significant expression in brain tissues versus other tissues (by t-test using log2-transformed data). Of the top 100 proteins highest in the s11 subtype, 58 were among the top 10% brain tissue-specific genes, a highly significant overlap ($p < 1E-35$, one-sided Fisher's exact test).

**Classification of external proteomic datasets**. We examined external, multi-cancer proteomic profiling datasets, classifying each external tumor or cell line profile by pan-cancer subtype as defined using our proteomic compendium dataset (see above). Within each cancer type of the external dataset being classified, we normalized log-transformed proteins to standard deviations from the median. As a classifier, we used the top set of proteins distinguishing between the pan-cancer subtypes. For each pan-cancer subtype, we computed the average normalized value for each protein, based on the centered expression data matrix. We then computed the Pearson correlation between each external profile and each pan-cancer subtype averaged profile. We assigned each external cancer case to a pan-cancer subtype, based on which subtype profile showed the highest correlation with the given external dataset profile. Supplementary Data 4 provides example calculations in Excel, by which the TCGA and CCLE proteomic profiles are classified according to proteome-based pan-cancer subtype. For the RPPA dataset, we used as a classifier the set of represented total protein features from which a significant association with a particular subtype was observable in the proteomic compendium dataset ($p < 0.001$ by t-test, based on logged and centered protein expression values). Where multiple RPPA features referred to the same protein, we randomly selected one feature.

**Gene signature analyses**. Our group previously collected gene signatures involving pathways of interest[21,23,57]. We computed pathway-associated gene expression signature scores (e.g., scores for EMT, hypoxia, KEGG: Glycolysis/Gluconeogenesis, KEGG: Pentose Phosphate pathway, KEGG: Fatty Acid metabolism, KEGG: TCA Cycle, and KEGG: Oxidative Phosphorylation or OX-PHOS, k-ras, MYC, Notch, NRF2/KEAP1, PI3K/mTOR, p53, Wnt, and YAP1) as follows, with all genes previously associated with the given signature used in the scoring calculation. We used the normalized values within the proteomic and transcriptomic compendium datasets (values normalized to standard deviations from the median within each dataset and cancer type). For NRF2/KEAP1, hypoxia, Notch, p53, KEGG, and GO term signatures, we computed the average expression of the set of genes within a given signature. For k-ras, MYC, PI3K/mTOR, Wnt, and YAP1 signatures, normalized expression profiles were scored for the above signatures using our t-score metric[58]. We generated gene signature scores of NRF2/KEAP1 pathway as described[59], based on four different signatures[57]. The hypoxia signature was the set of canonical HIF1A targets from Harris[60]. We generated gene transcription signature scores of YAP1 pathway, based on four different signatures[57]. The MYC signature (from data by ref. [61]) was from ref. [62]. The Settleman k-ras sensitivity signature was from ref. [63]. The Wnt signature was taken directly from ref. [64]. NOTCH signature was defined previously[65]. The PI3K/mTOR signature was defined in ref. [37], involving mRNAs modulated in vitro by inhibitors to PI3K or mTOR, according to CMap dataset ($p < 0.01$, comparing PI3K/mTOR-inhibited cells with the rest of the Cmap profiles). While we had previously developed PI3K/AKT/mTOR proteomic signatures, based on phosphorylation levels of canonical pathway members as measured by reverse-phase protein arrays[37], these proteins were not measured consistently in sufficient numbers of tumors across our mass spectrometry-based proteomic compendium. Gene targets of p53 were from ref. [66]. Scores for signatures of canonical nueroendocrine (NET) and epithelial-mesenchymal transition markers were computed as previously described[23]. Gene signatures for complement activation pathway, MMPs, collagen VI, hemoglobin complex, endoplasmic reticulum, and steroid biosynthetic pathway were defined using GO. Canonical markers of neuroendocrine tumors considered were CDX2, CHGA, ENO2, NCAM1, and SYP[23].

To computationally infer the infiltration level of specific immune cell types using proteomic data, we used a set of 501 genes specifically over-expressed in one of 24 immune cell types from ref. [67]. For each Bindea immune cell type signature, we computed the average expression of the set of genes within the given signature. For T cells, we computed a signature score based on the average normalized values of five canonical markers (LCK, CD3E, CD3D, CD3G, CD2). For cytotoxic T cells, we computed a signature score based on canonical markers CD8A and CD8B. For T helper and regulatory T cells, we considered CD4 marker expression. For an immune checkpoint pathway signature, we computed the average of normalized protein values for a set of previously defined genes representing targets for immunotherapy[57], including PDCD1 (PD1), CD247 (CD3), PDCD1LG2 (PDL2), CTLA4 (CD152), TNFRSF9 (CD137), and TNFRSF4 (CD134). In addition, sample profiles were scored for average normalized expression of Antigen Presentation MHC class I (APM1) genes (HLA-A/B/C, B2M, TAP1/2, TAPBP) and for average normalized expression of Antigen Presentation MHC class II (APM2) genes.

**Pathway-level somatic alteration categories**. For the pathway-centric view of somatic alterations across the proteomic compendium tumors (Fig. 7), we focused on key pathways and genes previously annotated across multiple cancer types based on domain knowledge (see Methods)[23,25,57,68]. Of the 2002 human tumors with proteomics data in our compendium, 1597 tumors had combined whole-exome and CNA data. Key pathways and genes considered included: RTK pathway (BRAF, EGFR, ERBB2, ERBB3, ERBB4, FGFR1, FGFR2, FGFR3, FGFR4, HRAS, KIT, KRAS, MET, NF1, NRAS), Hippo pathway (NF2, SAV1, WWC1), chromatin modification (CREBBP, EHMT1, EHMT2, EP300, EZH1, EZH2, KAT2A, KAT2B, KDM1A, KDM1B, KDM4A, KDM4B, KDM5A, KDM5B, KDM5C, KDM6A, KDM6B, KMT2A, KMT2B, KMT2C, KMT2D, KMT2E, NSD1, SETD2, SMYD4, SRCAP), SWI/SNF complex (ACTB, ACTL6A, ACTL6B, ARID1A, ARID1B, ARID2, BCL11A, BCL11B, BCL6, BCL6B, BRD7, BRD9, DPF1, DPF2, DPF3, PBRM1, PHF10, SMARCA2, SMARCA4, SMARCB1, SMARCC1, SMARCC2, SMARCD1, SMARCD2, SMARCD3, SMARCE1), mTOR pathway (AKT1, AKT2, AKT3, MTOR, PIK3CA, PIK3R1, PTEN, RHEB, STK11, TSC1, TSC2, IDH1, IDH2, VHL), MYC family (MYC, MYCN), Wnt/beta-catenin (APC, AXIN1, CTNNB1, FGF19, NCOR1), p53/Rb-related (ATM, CCND1, CCNE1, CDK4, CDKN1A, CDKN2A, E2F2, E2F3, FBXW7, MDM2, RB1, TP53), and NRF2 pathway (NFE2L2, KEAP1, CUL3, SIRT1, FH). For known oncogenes with hotspot mutations (e.g., AKT1, MTOR, PIK3CA, RHEB, BRAF, EGFR, ERBB2, ERBB3, FGFR2, HRAS, KRAS, NRAS, NFE2L2), if an SNV occurred in "hotspot" residues as reported by ref. [54], we considered the SNV in the analysis. At both the gene and pathway levels, we tabulated somatic alterations in the following order: SNV or indel, homozygous copy loss, heterozygous copy loss, high-level amplification (approximating five or more copies), copy gain (approximating 3–4 copies). WE considered heterozygous copy loss events for pathways RTK, Hippo, chromatin modification, SWI/SNF, mTOR, Wnt/beta-catenin, and p53/Rb-related. For MYC/MYCN, we considered copy gain in addition to high-level amplification. We did not include silent or synonymous SNVs in cataloging somatic alterations.

**Oncogene negative enrichment analyses**. TCGA datasets were compiled previously, with tumors assigned pathway-associated gene signature scores based on RNA-seq data and annotated according to pathway-centric somatic alterations as described above[23]. For TCGA tumors, we obtained somatic mutation calls from the publicly available "MC3" TCGA MAF file [https://www.synapse.org/#!Synapse:syn7214402]; variants called by two or more algorithms were used in this study. Analyses focused on the 10,224 tumors with RNA-seq data.

For both the proteomic compendium and TCGA datasets, we sought to identify enrichment patterns of small mutations within tumors that showed high pathway-associated expression signature but without a canonical alteration identified by pathway annotation. For this analysis, we focused on the 190 genes in both the COSMIC Cancer Consensus Gene List[39] and the set of significantly mutated genes from the TCGA PanCanAtlas project[40]. We evaluated each pathway for which both mutation annotation and expression signature scores were available (MYC/MYCN, Hippo, mTOR, NRF2, p53/Rb-related, RTK, Wnt/beta-catenin). Taking the set of tumors with high signature scoring for a given pathway (SD > 0.5 from the sample median) but with no canonical mutations (based on Fig. 7), we looked for statistical enrichment of SNV or indel events (by one-sided Fisher's exact test) involving any of the 190 genes. We compared results between the proteomic compendium and TCGA, and we selected for downstream analyses the genes with enrichment of mutation events for both datasets ($p < 0.01$ and $p < 0.001$, respectively, one-sided Fisher's exact test).

**Statistical analysis**. All $p$ values were two-sided unless otherwise specified. Nominal $p$ values do not involve multiple comparison adjustments, while FDRs involve $p$ values adjusted for multiple gene feature comparisons. We performed all tests using log2-transformed expression values. Visualization using heat maps was performed using both JavaTreeview (version 1.1.6r4)[69] and matrix2png (version 1.2.1)[70]. Figures indicate the exact value of $n$ (number of tumors), and the statistical tests used are noted in the figure legends and next to reported $p$ values in the Results section. Box plots represent 5% (lower whisker), 25% (lower box), 50% (median), 75% (upper box), and 95% (upper whisker).

**Reporting summary**. Further information on research design is available in the Nature Research Reporting Summary linked to this article.

## Data availability

All data used in this study are publicly available. Proteomics data are available for CPTAC and ICPC studies at the Proteomic Data Commons (https://pdc.cancer.gov/). For CPTAC studies, transcriptome data, copy number data, and small somatic mutation data are available at the Genomic Data Commons (https://gdc.cancer.gov/). Raw genomic and transcriptomic CPTAC data can be accessed via dbGap Study Accession: phs001287.v13.p5. Raw data may be obtained once authorized access is granted via Data Use Certification (DUC) agreement. Genomic and transcriptomic liver cancer data are available at https://www.biosino.org/node/project/detail/OEP000321. Genomic and transcriptomic prostate cancer data are available at found on European Genome-Phenome Archive (EGA), under accession EGA: EGAS00001000900. Genomic and transcriptomic gastric cancer data are available in NCBI SRA (PRJNA505380) and GEO (GEO: GSE122401) databases, respectively. TCGA RNA-seq data are also available from the Broad Institute's Firehose data portal (https://gdac.broadinstitute.org). The TCGA RPPA dataset is available from the TCPA portal (http://tcpaportal.org/tcpa/). Cancer Cell Line Encyclopedia (CCLE) datasets are available from the CCLE website (http://www.broadinstitute.org/ccle). For other published studies, molecular data availability information is provided in the associated publication. The compendium datasets of molecular profiles for total protein, phospho-protein, and mRNA—compiled as part of our study—are available through GitHub (https://github.com/chadcreighton/cancer-proteomics-compendium-n2002). Each molecular dataset is uploaded on GitHub as a series of separate project files, using a common protein feature set with the same ordering across files. One can concatenate the files together. Each molecular dataset has a common sample set, allowing one to derive correlations between datasets (e.g., between mRNA and protein). The phospho-protein datasets consist of 5419 phospho-protein features that had available data for >50% of samples in at least seven cancer types. Any remaining data are available within the Article or Supplementary Information. Source data are provided with this paper.

## Code availability

Source code in R (written using version 4.0.3) for identifying de novo pan-cancer proteomic subtypes, using ConsensusClusterPlus method (version 1.59.0), with an example data table of proteomic expression for the top ~2000 variable genes across 2002 tumor, is available at Github (https://github.com/chadcreighton/cancer-proteomics-compendium-n2002).

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

## Acknowledgements

This work was supported by National Institutes of Health (NIH) grant P30CA125123 (C.J.C.) and by NIH grant U54 CA118948 and Department of Defense (DOD) grant-W81XWH-19-1-0588 (S.V.).

## Author contributions

Conceptualization: C.J.C.; Methodology: C.J.C. and F.C.; Investigation: C.J.C., Y.Z., and F.C.; Formal analysis: C.J.C., Y.Z., F.C., and D.S.C.; Data curation: C.J.C., S.V., and

D.S.C.; Visualization; C.J.C. and Y.Z.; Writing: C.J.C.; Manuscript review: S.V., Y.Z., F.C., and D.S.C.; Supervision: C.J.C. and S.V.

## Competing interests

The authors declare no competing interests.
