## [Peer Review File · Nature Communications]

Proteogenomic characterization of 2002 human cancers reveals pan-cancer molecular subtypes and associated pathwaysREVIEWER COMMENTS

Reviewer #1 (Remarks to the Author): Expert in cancer genomics

NCOMMS-21-48038

Zhang et al. "Proteogenomic characterization of more than 2000 human cancers reveals pan-cancer molecular subtypes and associated pathways "

The authors continue their studies of computational biology analyses of proteomics and genomics data on human tumors that was begun in their ref 21 2019 Nature Communication paper where they analyzed some 500 human tumors for proteomics and transcriptomics using a very similar approach. In the current study they now analyze 2002 primary tumors from 14 cancer lineages and identify 11 proteomic based subtypes that, by and large, cut across several tumor lineages. Previously they had identified 10 subtypes and it appears that 70% plus of the subtypes identified in 2019 publication are found in the current analyses. They find some new subtypes enriched in brain tumors, and some tumors with Myc pathway related upregulation. Similarly to 2019 they deposit their findings in a UALCAN data portal. In the current paper they identify a proteomic genomic signatures associated with "aggressive" cancers across lineages and this appears directly related to their reference 22 which is Oncogene 2021 publication ("Mass-spectrometry-based proteomic correlates of grade and stage reveal pathways and kinases associated with aggressive human cancers"). No validation of their findings either with proteomics or immune histochemistry (IHC) in independent tumor samples is provided, and there are no functional tests of any of their findings to know if they are biologically relevant.

Comments to the Authors:

This article is reviewed in the context of both the urgent need to clinically and/or functional translate recent proteo-genomic findings of human tumors and for these huge new data sets to help guide mechanistic and new target discovery research to provide new insight into tumor pathogenesis and hopefully, early diagnosis, prevention and treatment. The paper is also reviewed in the context of multiple other papers analyzing portions of the deposited datasets including the authors own previous Nature Communication paper in 2019. The computational work appears technically well done and is presented in a descriptive style. The deposit of the datasets in UALCAN will provide for easy access and further analyses by other investigators. There are a series of issues that would have greatly enhanced the value of this report

1. Obviously, besides having more tumors and several other tumor types to analyze what do we know differently between the current report and the very similar report in Nature Communication in 2019? While there are some new findings, it is not clear to me that any of these are very impactful. If the authors feels some are, they need to clearly identify those so we can assess their potential impact.
2. The report in Oncogene 2021 is also very similar to the analyses of aggressive tumors in the current report. What are the differences between these, and how and why does do the analyses in current manuscript extend our understanding and approach to the problem of aggressive tumors?
3. What are the findings in the current report that provide new understanding of cancer biology, pathogenesis, diagnosis, or new treatment targets? Which of the new protein markers are important for clinical characterization and can these be analyzed using IHC? How does the current report advance our knowledge and does it provide a "roadmap" for future systematic analyses and laboratory evaluation? If so exactly what specifically is in this roadmap?
4. Are there new therapeutic targets identified and are these in some or multiple tumor lineages?
5. How do these data provide new mechanistic insights to tumor biology and behavior?
6. Which of the proteomic findings are in the tumor cells and which are in the tumor microenvironment? How are tumor and tumor microenvironmental proteomics changes related?
7. Does the current analyses give us any new mechanistic understanding for key driver oncogenes such as oncogenic KRAS or tumor suppressor genes such as TP53? What are the number and types of the proteomic signatures associated with mutated oncogenes like KRAS and how do these relate across lineages? Do any of these provide a therapeutic target?
8. What are the proteomic genomic findings that can help us relate tumor samples to human preclinical models such as tumor cell lines and patient derived xenografts (PDXs)? Specifically are

the proteomic genomic signatures from tumors that we could use to identify tumor lines and PDXs that closely correlate with the 11 different groups described in this manuscript?

9. Because there were multiple computational steps involved in the current analyses it would be important and useful to have a document like "Sweave" (<https://en.wikipedia.org/wiki/Sweave>) that would allow other investigators to independently and easily verify their findings.

10. Most importantly, this manuscript would have benefited by some type of independent validation of the findings such as IHC studies of an independent set of molecularly annotated tumors to verify several of the markers and at least some of the 11 subsets. Of course, this would not need to be on the scale of the 2000 tumors analyzed here, but the obvious thing to do would be to pick out a subset of proteins for which good antibodies and IHC methodologies exist and then show that studies (even of one or two lineages) could readily identify these subsets outside of the proteogenomics analyses.

Reviewer #2 (Remarks to the Author): Expert in MS-based proteomics

The manuscript "Proteogenomic characterization of more than 2000 human cancers reveals pan-cancer molecular subtypes and associated pathways" by Zhang et al., presents reanalysis of publicly available proteomics, transcriptomics, gene level copy number alteration and mutation dataset. Reanalysis was conducted with proper measures to tackle variability between different datasets. Finding of this study will further benefit the cancer research, manuscript is well written and I find it to be suitable for publication in its present form.

Reviewer #3 (Remarks to the Author): Expert in cancer proteogenomics

Zhang and co-workers describe findings made in a proteogenomic analysis of 2000 tumors from 14 different tumor entities. The respective data from 17 studies have already been published individually. Here, the authors were out to find commonalities and specifics across entities. Furthermore, they wanted to explore connections between genomic alterations (SNV, CNA) and total as well as phospho-protein levels. They identified 11 proteome-based subtypes that partially matched with tumor locations (e.g., brain) or molecular subtypes of particular entities (e.g., basal breast cancer). However, several of the proteome-based subtypes were observed in a number of different entities, potentially hinting at common driver mechanisms in these tumors. The findings could have implications for therapy decisions, if re-identified also in individual patients.

This is a very interesting study, presenting the initial characterization of a huge dataset. The observations and the accessibility of the underlying data will certainly contribute to give this high impact.

Comments:

Lines 85ff.: A thorough description of the tumor entities, including numbers, is given in the materials and methods section (compare lines 352ff.). The text in the results part should be shortened and mostly refer to that in the materials and methods.

Lines 114ff./lines 207ff.: The authors describe the correlation, or the lack of it, with CNA burden in two places of the results section. This should be harmonized by collating the text at either place. The authors should consider referring to previous work (e.g., PMID:24071851) where the co-occurrence of oncogenic signatures had been described in a pan-cancer approach. In that paper, also the correlation between TP53 and CNAs was described.

Lines 187 and 189: The authors should remove other redundancies (here: the notion that s11 tumors comprise of all brain tumors).

Lines 245ff.: The authors found the p53 signature to being exceptional as only this signature was not different between somatically altered versus unaltered cases. The authors might consider discussing this a little (compare also my comment to lines 114/207). For example, they could add a statement on the different activities p53 has in the respective cellular context, when they

already do discuss about the lack of p53 pathway-associated mutations with protein data (lines 299ff.). Such statement might remain tumor cell centric (e.g., PMID:2832726, PMID:8023157), however, could expand also to other cells in the stroma (e.g., PMID:29866855).

In general, the authors should go beyond the sheer description of their observations in one or two places. Along these lines, they should try to connect some biology, placing some of their observations into biological context. For example, why is subtype s11 specifically detected in brain tumors (what are the likely brain-specific proteins that define this subtype)? In general, the authors should try to connect their findings with previous literature (e.g., on the role of Wnt-signaling in colorectal cancer and other entities). Then, they would well go beyond their statement in lines 306f.: '... would suggest that true biology is involved here.'

Lines 256ff.: The authors found phosphorylation sites in 58 of 106 phospho-proteins that had not been annotated before (in HPRD). Are the four sites (line 263) in PTEN, ERBB2 and another unspecified protein the only ones? How many other and which phosphosites were significantly correlated with pathway-level somatic variants or pathway-level gene signatures? Such correlation would render the respective phosphosites particularly interesting to follow up on.

Line 446: The authors identified k=2 to k=20 subtypes in unsupervised clustering. While they had used the term 'k' in a previous paper, they mostly use the term 's' in the current study.

Our responses and comments are marked in red italics.

Reviewer #1: Expert in cancer genomics

The authors continue their studies of computational biology analyses of proteomics and genomics data on human tumors that was begun in their ref 21 2019 Nature Communication paper where they analyzed some 500 human tumors for proteomics and transcriptomics using a very similar approach. In the current study they now analyze 2002 primary tumors from 14 cancer lineages and identify 11 proteomic based subtypes that, by and large, cut across several tumor lineages. Previously they had identified 10 subtypes and it appears that 70% plus of the subtypes identified in 2019 publication are found in the current analyses. They find some new subtypes enriched in brain tumors, and some tumors with Myc pathway related upregulation. Similarly to 2019 they deposit their findings in a UALCAN data portal. In the current paper they identify a proteomic genomic signatures associated with “aggressive” cancers across lineages and this appears directly related to their reference 22 which is Oncogene 2021 publication (“Mass-spectrometry-based proteomic correlates of grade and stage reveal pathways and kinases associated with aggressive human cancers”). No validation of their findings either with proteomics or immune histochemistry (ICH) in independent tumor samples is provided, and there are no functional tests of any of their findings to know if they are biologically relevant.

This article is reviewed in the context of both the urgent need to clinically and/or functional translate recent proteo-genomic findings of human tumors and for these huge new data sets to help guide mechanistic and new target discovery research to provide new insight into tumor pathogenesis and hopefully, early diagnosis, prevention and treatment. The paper is also reviewed in the context of multiple other papers analyzing portions of the deposited datasets including the authors own previous Nature Communication paper in 2019. The computational work appears technically well done and is presented in a descriptive style. The deposit of the datasets in UALCAN will provide for easy access and further analyses by other investigators. There are a series of issues that would have greatly enhanced the value of this report

We thank the reviewer for evaluating our work. Below we address the specific concerns raised.

1. Obviously, besides having more tumors and several other tumor types to analyze what do we know differently between the current report and the very similar report in Nature Communication in 2019? While there are some new findings, it is not clear to me that any of these are very impactful. If the authors feels some are, they need to clearly identify those so we can assess their potential impact.

We have expanded the Discussion section in light of these comments. While the present study does validate several observations regarding proteome-based subtypes that were made previously in our Nature Communications 2019 paper (PMID: 31831737), extending the initial observations to additional cancer types, there is much, much more to the present study that previous studies have not covered. A word to keep in mind is “proteogenomics,” the integration of proteome with genomic information. Our two previous cancer proteomics studies were not proteogenomics studies. Only the present study can have “proteogenomics” in the title. Beyond the additional tumors and cancer types represented in our proteomic compendium of 2002 tumors, the pan-cancer datasets compiled in the present study involve multiple levels of molecular data in addition to the proteome, including mRNA-level and DNA-level data

on the same set of tumors. In contrast, due in part to the timing of data releases, our first pan-cancer proteomic tumor subtyping study focused mainly on proteomic data in CPTAC tumors, and our recent study examining tumor grade correlates (PMID: 33627787) incorporated both protein and mRNA data but not somatic DNA-level data on CPTAC tumors. Our present proteogenomics survey enabled us to explore the interplay between genome and proteome, including aspects not represented in the transcriptome. In the present study, all figures after the Figure 3 de novo proteome-base subtyping (namely, Figures 5-10) utilize somatic genomic information in conjunction with proteomic data on a common set of tumors, representing analyses not carried out in our previous studies. This present study uses the largest collection of mass spectrometry-based proteomics on human tumors, which is also important for the proteogenomic analyses, as tumor subset analyses according to genomic status (e.g., pathway altered versus unaltered) are carried out. If we had attempted these types of analyses in our previous 2019 study, we would have been underpowered in many instances. Being limited to 300-500 tumors in total may leave some tumor subsets with too few cases to enable us to identify robust differences involving sparse mutation events.

Beyond the various proteogenomic analyses, some aspects of our present study covered ground explored in our two previous cancer proteomic studies, extending the observations to additional cancer types. These findings also have value and, for example, would add to the resource aspect of the present study (something we describe further under comment #3). We also uncovered additional subtypes in the present study not previously reported. We find many of the same pathways uncovered previously for the correlation by grade analyses, but this could also be considered an important finding. Such a finding implies that protein correlates of aggressive cancer would converge upon a common set of pathways, even if the individual cancer types would have distinct sets of protein correlates from each other. With this unique proteomic compendium we have assembled, with its various omics levels, we wanted to cover as much ground as possible for one strong and comprehensive paper, rather than divvying up the various results into two or more lesser papers.

2. The report in Oncogene 2021 is also very similar to the analyses of aggressive tumors in the current report. What are the differences between these, and how and why does do the analyses in current manuscript extend our understanding and approach to the problem of aggressive tumors?

As we note in the revised manuscript, the set of top enriched pathways uncovered here overlapped highly with the previously uncovered pathways based on five of the nine cancer types represented in our compendium. This finding suggests a core set of pathways associated with more aggressive tumors, even as the proteomic grade correlates differ by cancer type. We would consider this a finding. Additional pathways in this study include alpha 6 beta 4 signaling (liver, pancreas) and oxidative damage (prostate, uterine). In addition, the proteomic grade correlates for the cancer types not analyzed previously would add to the resource value of our present study. The corresponding grade-level proteomic correlations would be pertinent for researchers who may study the four additional cancer types not represented previously (liver, lung squamous, pancreas, prostate).

3. What are the findings in the current report that provide new understanding of cancer biology, pathogenesis, diagnosis, or new treatment targets? Which of the new protein markers are important for clinical characterization and can these be analyzed using IHC? How does the current report advance our

knowledge and does it provide a “roadmap” for future systematic analyses and laboratory evaluation? If so exactly what specifically is in this roadmap?

We have expanded the Discussion section in light of these comments. As originally conceived and implemented, our study is a “landscape” study to increase understanding as to the molecular underpinnings of cancers beyond tissue-oriented domains, similar in approach to the marker papers from TCGA or CPTAC and to our previous pan-cancer subtyping studies (PMID: 29440175, PMID: 31831737). Knowledge of the molecular landscape of cancer may eventually translate to new therapeutic strategies in ways we may not anticipate at present. But in the meantime, knowing, for example, that most cancers may fall into a distinct set of categories based on the proteome, or demonstrating how a somatic genomic alteration may translate to changes in the proteome, are important first steps.

Our study results provide an excellent framework for understanding the molecular landscape of cancers at the proteome level. By proteogenomics analysis, we found that somatic alterations of cancer-associated pathways are reflected in the cancer proteome, whereby tumors with somatic alteration involving genes in a pathway tend to show higher levels of protein-based signature scoring for that pathway. The proteomic and genomic data integration represents orthogonal information pointing to a common tumor subset deregulated for a given pathway. Our collective knowledge of molecular pathways has been largely derived from experimental models. The signature analyses allow us to explore experimentally inferred cause-and-effect relationships in the human disease setting, whereby these relationships manifest in tumors as significant correlations. Pathway activity, as measured using gene signatures applied to proteome data, reflects known mutations or copy alteration in most but not all tumors examined, suggesting additional, unexplained, or underappreciated mechanisms of pathway activation. In the case of the MYC pathway, we identified several genes, many of which would have an under-appreciated role in the MYC pathway, for which somatic mutation was associated with higher pathway activation.

The associated datasets and gene-level associations represent a resource for the research community, including helping to identify gene candidates for functional studies. When previously surveying proteomic correlates of tumor grade, we identified specific protein kinases having a functional impact in vitro in uterine endometrial cancer cells, which provides a template for other researchers to utilize the gene-level associations provided in the present study. Regarding the pan-cancer proteome-based subtypes, a number of these appear manifested in cancer cell lines (new Figure 4), and additional data, such as DepMap gene dependency data, may be leveraged to help select targets of interest to examine in these cell lines. RPPA profiling data can classify tumors according to our proteome-based subtypes, where RPPA antibody-based features may also lend themselves to relevant immunohistochemistry studies (with the RPPA antibody information being available as part of Data File S4). We have added the proteomic datasets to the user-friendly UALCAN data portal for all cancer types studied, facilitating differential analyses by protein and giving the research community ready access to our results. Our UALCAN data portal has been heavily accessed and utilized over the past few years, with over 810,000 page views and over 2400 citations to date of our initial paper describing UALCAN (PMID: 28732212). All the protein-level and mRNA-level associations described in our study are available in the Supplementary Data files as described, providing researchers another level of access to our results.

Regarding potential future IHC studies, to the revised Data File S2, we have added relevant data from the Human Protein Atlas (proteintlas.org), including antibody validation information for immunohistochemistry and immunocytochemistry/IF.

4. Are there new therapeutic targets identified and are these in some or multiple tumor lineages?

In the revised Data File S4 (featuring the protein-level subtype associations), we have provided for each protein any association in the DrugBank database (PMID: 24203711), a comprehensive online database containing extensive information about any drugs targeting specific genes. In the Results section, we note that of the 1073 proteins for which total levels best distinguished the subtypes, 225 proteins had a drug target association by DrugBank. We provide these 225 proteins in a separate Excel tab in Data File S4, which proteins would span multiple tumor lineages. Similarly, the Drugbank associations can help filter for other protein-level associations of interest provided in Data File S2.

5. How do these data provide new mechanistic insights to tumor biology and behavior?

The new Figures 4 and 8 of the manuscript revision—described in detail under our responses to reviewer comments #8 and #7, respectively—would regard the application of our results in a mechanistic manner. Additional findings with mechanistic implications involve Figure 7, which shows that somatic alterations of cancer-associated pathways are reflected in the cancer proteome, whereby tumors with somatic alteration involving genes in a pathway tend to show higher levels of protein-based signature scoring for that pathway. Our collective knowledge of molecular pathways has been largely derived from experimental models. The signature analyses allow us to explore experimentally inferred cause-and-effect relationships in the human disease setting, whereby these relationships manifest in tumors as significant correlations. When previously surveying proteomic correlates of tumor grade, we identified specific protein kinases having functional impact in vitro in uterine endometrial cancer cells. That approach provides a template for other researchers to utilize the gene-level associations provided in the present study (both via the supplementary data files and the UALCAN data portal)..

6. Which of the proteomic findings are in the tumor cells and which are in the tumor microenvironment? How are tumor and tumor microenvironmental proteomics changes related?

As we note in the manuscript, we observed multiple distinct subtypes involving the immune system, one involving the adaptive immune response and T-cell activation (s2 subtype), and others associated with the humoral immune response and complement pathway (s5 and s6 subtypes). We also observed tumor stroma-associated subtypes (s5 and s6 subtypes), one subtype involving collagen VI network (s6 subtype).

In the revised manuscript, we classified 378 cancer cell lines with mass-spectrometry-based proteomic data according to s1-s11 subtypes, with the analyses presented in the new Figure 4 and described in detail under our response to comment #8. Consistent with our previous results (PMID: 29440175, PMID: 31831737), we note in the revision that not all differential patterns observable in human tumor proteomic data appeared as strong in the cell line proteomic data, particularly with regards to immune-associated or stroma-associated subtypes, which is attributable to various factors including growth conditions of cell lines lacking tumor microenvironmental effects.

7. Does the current analyses give us any new mechanistic understanding for key driver oncogenes such as oncogenic KRAS or tumor suppressor genes such as TP53? What are the number and types of the

proteomic signatures associated with mutated oncogenes like KRAS and how do these relate across lineages? Do any of these provide a therapeutic target?

In the revised manuscript, we have added another main figure—Figure 8, also shown and described below—which features proteomic signatures involving mutant TP53 or mutant KRAS. Figure 8a (also shown below) shows differential expression heat maps of top total protein features correlated with small somatic mutation of TP53 (left, FDR<10%) or with somatic hotspot mutation of KRAS (right, FDR<10%) across all cancers studied (n=1440 tumors with protein and mutation data).

The TP53 and KRAS mutation signatures each span multiple tumor lineages. As expected, KRAS mutation is common in specific cancer types (colorectal, lung adenocarcinoma, pancreas, uterine). As the proteomic profiles in our proteomic compendium dataset are centered within each cancer type, cancer type differences are not a factor here. The genes presented above were also significant in a regression model that factored cancer type as a covariate.

Interestingly, a substantial number of proteins differentially expressed with TP53 mutation were not reflected at the mRNA level (Figure 8b, also shown below). Out of 2752 proteins differential with TP53 mutation (FDR<10%), 1134 (41%) were not similarly altered significantly ($p < 0.05$) at the mRNA level (Data Files S2 and S3). Similarly, out of 5414 genes with mRNA differential with TP53 mutation (FDR<10%), 2272 (42%) were not similarly altered at the protein level.

TP53 mutation associates with higher overall mutational burdens in cancer, and most of the proteomic signature of TP53 mutation overlapped with the proteomic signature of CNA burden from Figure 6; at the same time, 638 (23%) of the 2752 TP53 mutation-associated genes (FDR<10%) were not significant in the same direction ($p < 0.05$) in the CNA burden analysis (Data File S2). The KRAS mutant gene signatures involved 199 proteins and 580 mRNAs (FDR<10%), with 109 of the significant proteins

showing the same trend ($p < 0.05$) at the mRNA level. Genes with expression higher with TP53 mutation at both the protein and mRNA levels included genes involved in cell division, while genes higher specifically at the protein level involved SWI/SNF complex, and genes higher specifically at the mRNA level involved the proteasome, NF-kappaB signaling, and Wnt pathway (Figure 8c, also shown below). Genes with expression higher or lower with KRAS mutation at the mRNA level included genes involved in cell adhesion or humoral immune response, respectively (Figure 8c, also shown below).

Individual genes of interest that were associated with TP53 or KRAS mutation at the protein but not mRNA levels would have known functional roles with the respective pathways (associations for all genes provided in Data Files S2 and S3). For example, protein levels for p53 and HEATR1 were higher with TP53 mutation (new Figure 8d, also shown below), where depletion of HEATR1 would lead to impaired proliferation and induction of p53-dependent cell cycle arrest (PMID: 29143558). Proteins higher with KRAS mutation included INSR (insulin receptor)—where insulin can promote invasion and migration of KRAS mutant cells (PMID: 30838710)—and ERBIN (erb2 interacting protein)—which can regulate KRAS mutant-induced tumorigenesis (PMID: 26056141).

Also, when combining the set of proteins higher in KRAS mutant tumors (FDR<10%) with the set of genes with DrugBank entries (Data Files S2 and S4), we get 27 proteins, including EPHA2, EPHB4, INSR, and PGK1.

In addition, as part of this study, we have implemented a new feature in UALCAN, which allows the user to compare differences for a given protein by pathway somatic alteration status across cancer types. An example plot from this feature is provided below.

<http://ualcan.path.uab.edu/cgi-bin/Pan-cancer-CPTAC1.pl?genenam=HEATR1>

8. What are the proteomic genomic findings that can help us relate tumor samples to human preclinical models such as tumor cell lines and patient derived xenografts (PDXs)? Specifically are the proteomic genomic signatures from tumors that we could use to identify tumor lines and PDXs that closely correlate with the 11 different groups described in this manuscript?

In the revision, we have added a new main figure (Figure 4), which examined proteomic datasets external to our proteomic compendium dataset. These external datasets include the Cancer Cell Line Encyclopedia (CCLE), for which mass spectrometry-based proteomic data have been generated (PMID: 31978347). Using a protein-based classifier developed from the proteomics compendium dataset, we classified 375 cancer cell lines with mass-spectrometry-based proteomic data according to s1-s11 subtypes (Data File S4 and Figure 4c, also shown below, 378 profiles representing 375 cell lines).

Given the reviewer's comments regarding functional tests (e.g., the introductory comments above), we took the opportunity to integrate functional data on the CCLE cell lines with the CCLE subtyping results. For 301 of the 375 cell lines, CRISPR knockout screens globally assessed gene essentiality (PMID: 34930405). In taking the global correlation between differential protein expression profile versus gene effect scoring profile for each cell line, s3 and s11 cell lines had consistent negative correlations in contrast to the other subtypes (Figure 4d, also shown below). This observation indicated that s3 and s11 cell lines (and by extension their tumor counterparts) tended to express essential genes highly.

For s3 and s11 cell lines, Figure 4e (also shown below) represents the associated patterns for the top sets of genes having both high expression and low gene effect scores across cell lines (with at least half of cell lines having normalized expression > 0.5SD and gene effect scores < -0.5).

In summary, regarding the pan-cancer proteome-based subtypes, a number of these appear manifested in cancer cell lines, and additional data, such as DepMap gene dependency data, may be leveraged to help select targets of interest to examine in these cell lines. Supplementary Data File S4 provides the CCLE cell line assignments to proteome-based subtype, along with the underlying proteomic data.

9. Because there were multiple computational steps involved in the current analyses it would be important and useful to have a document like “Sweave” (<https://en.wikipedia.org/wiki/Sweave>) that would allow other investigators to independently and easily verify their findings.

We have taken additional steps to make the underlying datasets readily accessible to everyone in the manuscript revision. All data used in our study are publicly available. Therefore, in principle, others could go to the original data sources described in the Data Availability section and independently assemble the proteomic and transcriptomic datasets for the 2002 tumors examined in our study. However, this would involve a great deal of work, equal to the considerable efforts spent by our group, which represents a barrier for anyone wishing to follow up on our study. In response to the reviewer’s comment, we gave some thought as to how to make the proteomic compendium dataset available, as it represents a considerably large file (>300Mb), too large for inclusion in the journal’s supplemental section. We settled on GitHub and have posted the data at <https://github.com/chadcreighton/cancer-proteomics-compendium-n2002>. The compendium datasets of molecular profiles for total protein, phospho-protein, and mRNA—compiled as part of our study—are available through GitHub. Each molecular dataset is available on GitHub as a set of separate project files. Each file uses a common protein feature set with the same ordering across files. One can concatenate the files together, e.g., using Excel. Each molecular dataset has a common sample set, allowing one to derive correlations between datasets (e.g., between mRNA and protein). GitHub limits the size of files uploaded, which required us to break down the datasets according to the original studies.

Our study did not use any specialty source code. On GitHub (<https://github.com/chadcreighton/cancer-proteomics-compendium-n2002>), we have source code in R to identify de novo pan-cancer proteomic subtypes, using ConsensusClusterPlus method, with an example data table of proteomic expression for the top ~2000 variable genes across 2002 tumors. Regarding the classification of external tumor or cell line proteomic profiles by proteome-based subtype (Figure 4), Data File S4 provides the sets of protein features used to classify the external proteomic datasets (TCGA-RPPA, and CCLE-mass spectrometry). The actual calculations to assign subtype based on the classifier are provided in Excel.

The manuscript revision includes a Source Data file in Excel, according to Nature Communications specifications (<https://www.nature.com/ncomms/submit/how-to-submit>). The Source Data appears optional for the journal, but we have provided it to address the reviewer's comment further. The source data file either contains or points to the raw data underlying any graphs and charts presented in the figures. Each figure containing relevant data is represented by a single sheet in the Excel document as part of the source data file. For the most part, our Source Data points to the relevant Supplementary Data files for each figure component, as all gene-level associations used in the study have been made available in the Supplementary Excel files.

While we have made every effort here to provide the underlying data and results for our study to the community for reproducible research, we would not be able to provide an Sweave document to go with the various analyses performed in the study. If all analyses were performed in R or a similar program, that might lend itself to Sweave. Sweave would appear to lend itself to well-defined analyses for a smaller dataset but may not be well suited for large-scale integrative analyses, such as those of our present study. Across our entire study, our integrative analysis work required us to do a great deal of work in Excel (as well as in Adobe Illustrator and Java Treeview) in addition to R. For example, we carried out the collating of the individual datasets, including centering of the gene expression values within each cancer type, in Excel. We see one drawback to doing everything in R involving the need to be very precise in coding, where programmatic mistakes can occur (e.g., off by one errors), and one may not realize it until much later.

10. Most importantly, this manuscript would have benefited by some type of independent validation of the findings such as IHC studies of an independent set of molecularly annotated tumors to verify several of the markers and at least some of the 11 subsets. Of course, this would not need to be on the scale of the 2000 tumors analyzed here, but the obvious thing to do would be to pick out a subset of proteins for which good antibodies and IHC methodologies exist and then show that studies (even of one or two lineages) could readily identify these subsets outside of the proteogenomics analyses.

As an independent validation, we used the Reverse Phase Protein Array (RPPA) platform, which, like IHC, is antibody-based and independent of the mass spectrometry platform. In the revision, we have added a new main figure (Figure 4), which examined proteomic datasets external to our proteomic compendium dataset for evidence of the manifestation of our proteome-based pan-cancer subtypes. Using a protein-based classifier developed from the proteomics compendium dataset, we classified 7694 TCGA tumors with RPPA data according to s1-s11 proteome-based pan-cancer subtypes (Figure 4a, also shown below).

We had previously classified the TCGA RPPA profiles according to k1 through k10 subtypes (PMID: 31831737), and the relationships between the s1-s11 and k1-k10 subtype classifications in the TCGA RPPA dataset mirrored the relationships observed in the proteomic compendium dataset (Figures 4b and 3a, also shown below). Also consistent with the proteomic compendium results, TCGA s3 tumors were highly enriched for basal-like breast cancer ($p < 1E-35$, one-sided Fisher's exact test), TCGA s11 tumors were highly enriched for brain tumors (TCGA GBM and LGG projects, $p < 1E-50$), and TCGA s10 tumor were moderately enriched for brain tumors ($p < 0.005$). We can note that the association p -values for the TCGA-RPPA dataset are very small, given the large patient numbers involved (black on the right panel representing $p < 1E-80$).

proteomic compendium dataset results (Fig 3a)

TCGA-RPPA dataset results (Fig 4a)

In conclusion, the proteome-based subtypes defined using our proteomic compendium dataset could be observable in an independent cohort, using an orthogonal, antibody-based proteomics platform. RPPA profiling data can classify tumors according to subtype, where RPPA antibody-based features may also lend themselves to relevant immunohistochemistry (IHC) studies. Data File S4 provides the sets of protein

features used to classify the external TCGA-RPPA dataset. For these TCGA-RPPA protein features, RPPA antibody information (company, catalog number, species, validation status) is provided where available, many of which in principle could also be applied to future IHC studies. In addition, to the revised Data File S2, we have added relevant data from the Human Protein Atlas (proteinallas.org), including antibody validation information for immunohistochemistry and immunocytochemistry/IF.

Reviewer #2: Expert in MS-based proteomics

The manuscript “Proteogenomic characterization of more than 2000 human cancers reveals pan-cancer molecular subtypes and associated pathways” by Zhang et al., presents reanalysis of publicly available proteomics, transcriptomics, gene level copy number alteration and mutation dataset. Reanalysis was conducted with proper measures to tackle variability between different datasets. Finding of this study will further benefit the cancer research, manuscript is well written and I find it to be suitable for publication in its present form.

We thank the reviewer for evaluating our work.

Reviewer #3: Expert in cancer proteogenomics

Zhang and co-workers describe findings made in a proteogenomic analysis of 2000 tumors from 14 different tumor entities. The respective data from 17 studies have already been published individually. Here, the authors were out to find commonalities and specifics across entities. Furthermore, they wanted to explore connections between genomic alterations (SNV, CNA) and total as well as phospho-protein levels. They identified 11 proteome-based subtypes that partially matched with tumor locations (e.g., brain) or molecular subtypes of particular entities (e.g., basal breast cancer). However, several of the proteome-based subtypes were observed in a number of different entities, potentially hinting at common driver mechanisms in these tumors. The findings could have implications for therapy decisions, if re-identified also in individual patients.

This is a very interesting study, presenting the initial characterization of a huge dataset. The observations and the accessibility of the underlying data will certainly contribute to give this high impact.

We thank the reviewer for evaluating our work.

Comments:

Lines 85ff.: A thorough description of the tumor entities, including numbers, is given in the materials and methods section (compare lines 352ff.). The text in the results part should be shortened and mostly refer to that in the materials and methods.

We have opted to keep a brief description of the tumor entities and numbers at the beginning of the Results section while expanding on the description of the tumor cohort in Methods, pending any recommendations from the editorial team. We reason that most general readers do not read the Methods section when perusing papers, where methods are supposed to contain the nitty-gritty details that a subset of readers would want to know. But readers will want to know what tumor types and numbers are

involved in the study, and they should be able to readily have this information without having to browse through the figures or Methods.

Lines 114ff./lines 207ff.: The authors describe the correlation, or the lack of it, with CNA burden in two places of the results section. This should be harmonized by collating the text at either place. The authors should consider referring to previous work (e.g., PMID:24071851) where the co-occurrence of oncogenic signatures had been described in a pan-cancer approach. In that paper, also the correlation between TP53 and CNAs was described.

In the manuscript revision, we note the association of TP53 mutation with higher overall CNA burdens in cancer. In response to Reviewer #1's comment #7, the revision features an additional main figure (Figure 8) exploring proteomic and transcriptomic signatures associated with somatic mutation of TP53 (described in detail above). Perhaps not surprisingly, the proteomic signature of CNA burden overlaps considerably with the proteomic signature of TP53 mutation, which we are careful to note in the manuscript. At the same time, we also note that 638 (23%) of the 2752 TP53 mutation-associated genes (FDR<10%) were not significant in the same direction ($p<0.05$) in the CNA burden analysis. And so, there is some distinction between the two signatures. Data File S2 provides all the gene-level statistics for the CNA burden associations and the TP53 mutation associations.

Lines 187 and 189: The authors should remove other redundancies (here: the notion that s11 tumors comprise of all brain tumors).

We have opted to keep the redundancy for a couple of reasons. The first mention of s11 comprising all brain tumors has to do with both s4 and s11 being associated with neuroendocrine markers, whereas the second mention has to do with the section exploring the new s10 and s11 subtypes. So, there is a different context for each statement. In addition, the manuscript revision features an additional main figure (Figure 4) examining the subtypes in external datasets, which related text is between the original lines 187 and 189. When revisiting s11 in Figure 5, we want to remind the reader of the brain tumor association, rather than assuming that the reader has read and remembered all the details regarding the earlier section. Readers may enter a paper at different places, often not reading sequentially from Intro to Discussion.

Lines 245ff.: The authors found the p53 signature to being exceptional as only this signature was not different between somatically altered versus unaltered cases. The authors might consider discussing this a little (compare also my comment to lines 114/207). For example, they could add a statement on the different activities p53 has in the respective cellular context, when they already do discuss about the lack of p53 pathway-associated mutations with protein data (lines 299ff.). Such statement might remain tumor cell centric (e.g., PMID:2832726, PMID:8023157), however, could expand also to other cells in the stroma (e.g., PMID:29866855).

In the revised manuscript, we have added the following text to expand on the lack of correlation of the p53 proteomic signature with pathway-level somatic alteration: "The p53 signature consisted of 27 canonical p53 transcriptional targets[ref] represented in the proteomic compendium dataset. At the mRNA level, 17 of the 27 p53 signature genes had a significant negative expression correlation ($p<0.05$) with TP53 mutation status, while at the protein level, only 9 of the 27 genes had a similar negative correlation." In addition, in response to Reviewer #1's comment #7, the revision features an additional

main figure (Figure 8) exploring proteomic and transcriptomic signatures associated with somatic mutation of TP53 (described in detail above). While stroma-associated patterns are observable across tumors, including the stroma-associated molecular subtypes, the proteogenomic associations of p53-related somatic DNA alterations with p53 signature presumably have to do with the cancer cells specifically, as the stroma cells are not somatically altered. The stroma component might add some noise level to the data, but the proteogenomic associations we uncovered would be strong enough to overcome such noise.

In general, the authors should go beyond the sheer description of their observations in one or two places. Along these lines, they should try to connect some biology, placing some of their observations into biological context. For example, why is subtype s11 specifically detected in brain tumors (what are the likely brain-specific proteins that define this subtype)? In general, the authors should try to connect their findings with previous literature (e.g., on the role of Wnt-signaling in colorectal cancer and other entities). Then, they would well go beyond their statement in lines 306f.: ‘... would suggest that true biology is involved here.’

The revision reports that proteins high in s11 tumors were highly enriched for brain tissue-specific genes by GTEX (<https://www.gtexportal.org>). Taking the GTEX dataset, the 11311 genes with average TPM values greater than 5 and with Entrez ID represented in our proteomic compendium dataset were considered. From GTEX, we selected the top 10% of genes with the most significant expression in brain tissues versus other tissues (by t-test using log2-transformed data). Of the top 100 proteins highest in the s11 subtype, 58 were among the top 10% brain tissue-specific genes, a highly significant overlap ($p < 1E-35$, one-sided Fisher’s exact test).

Regarding the mutation associations by cancer type presented in Figure 7a and Figure s3, we have added the following to the revision: “Expected associations of cancer type with somatic pathway alteration were observed, including MYC amplification in breast cancer; Wnt pathway alterations via APC mutation in colorectal cancer; NRF2 pathway alteration in squamous lung cancer; KRAS mutations in colorectal, lung adenocarcinoma, and pancreatic cancers; TP53 mutations in ovarian lung squamous, and head and neck cancers; and mTOR pathway alterations via VHL mutation in renal cancer.”

Elsewhere in the manuscript, we cite the literature when highlighting genes of interest. For example, with the new analyses defining signatures of TP53 mutation or KRAS mutation, we cite papers involving the genes highlighted in Figure 8d. Also, regarding the novel genes associated with MYC signature (Figure 10), we find that many of these do have at least one publication supporting their role in MYC pathway, even if these roles have been underappreciated.

Lines 256ff.: The authors found phosphorylation sites in 58 of 106 phospho-proteins that had not been annotated before (in HPRD). Are the four sites (line 263) in PTEN, ERBB2 and another unspecified protein the only ones? How many other and which phosphosites were significantly correlated with pathway-level somatic variants or pathway-level gene signatures? Such correlation would render the respective phosphosites particularly interesting to follow up on.

The four sites were the only ones significantly correlated with both somatic variants and gene signatures. Data File S6 has the data underlying the 106 phospho-proteins, including those not found in HPRD. The revised text in the manuscript related to this is as follows: “Of the 106 phospho-proteins, 36 were

significantly correlated ($p < 0.05$, Pearson's) with the corresponding pathway-level DNA somatic mutation class (Figure 9a, 15 of these representing potentially novel phosphosites), and 24 were significantly correlated with the corresponding pathway-level gene signature (Figure 9b, 10 of these representing novel phosphosites), nine of which were significantly correlated with pathway-level mutation class. Of the nine phospho-proteins, four, including PI3K/AKT/mTOR-related PTEN:s294 and RTK-related ERBB2:t671 and ERBB2:s968, were not found in HPRD (Figure 9c)."

Line 446: The authors identified k=2 to k=20 subtypes in unsupervised clustering. While they had used the term 'k' in a previous paper, they mostly use the term 's' in the current study.

In the revised manuscript, we have added some text in Methods to clarify the use of the k notation, which comes from the "k-means" method used in ConsensusClusterPlus. "Our previous proteome-based molecular subtyping study[ref] had referred to its set of subtypes as k1 through k10, based on the 'k' in the k-means algorithm used in ConsensusClusterPlus. In this present study, we referred to the new set of subtypes of s1 through s11—the 's' representing 'subtype'—as we wanted to draw distinctions between the respective sets of cluster assignments."

REVIEWERS' COMMENTS

Reviewer #1 (Remarks to the Author):

The authors have responded appropriately to all of the reviewers' comments.

Reviewer #3 (Remarks to the Author):

the authors have substantially improved their manuscript and addressed all points I had.

Our responses and comments are marked in red italics.

Reviewer #1: Expert in cancer genomics

The authors have responded appropriately to all of the reviewers' comments.

We thank the reviewer for taking the time to evaluate our work.

Reviewer #3: Expert in cancer proteogenomics

The authors have substantially improved their manuscript and addressed all points I had.

We thank the reviewer for taking the time to evaluate our work.